# Perturbing local steroidogenesis to improve breast cancer immunity

Qiuchen Zhao[1,2,6], Jhuma Pramanik[1,6], Yongjin Lu[3], Natalie Z. M. Homer [4], Charlotte J. Imianowski [1], Baojie Zhang[1], Muhammad Iqbal[1], Sanu Korumadathil Shaji[1], Andrew Conway Morris [1], Rahul Roychoudhuri [1], Klaus Okkenhaug [1], Pengfei Qiu [3,5] ✉ & Bidesh Mahata [1] ✉

Breast cancer, particularly triple-negative breast cancer (TNBC), evades the body's immune defences, in part by cultivating an immunosuppressive tumour microenvironment. Here, we show that suppressing local steroidogenesis can augment anti-tumour immunity against TNBC. Through targeted metabolomics of steroids coupled with immunohistochemistry, we profiled the existence of immunosuppressive steroids in TNBC patient tumours and discerned the steroidogenic activity in immune-infiltrating regions. In mouse, genetic inhibition of immune cell steroidogenesis restricted TNBC tumour progression with a significant reduction in immunosuppressive components such as tumour associated macrophages. Steroidogenesis inhibition appears to bolster anti-tumour immune responses in dendritic and T cells by impeding glucocorticoid signalling. Undertaking metabolic modelling of the single-cell transcriptomics and targeted tumour-steroidomics, we pinpointed the predominant steroidogenic cells. Inhibiting steroidogenesis pharmacologically using a identified drug, posaconazole, curtailed tumour expansion in a humanised TNBC mouse model. This investigation paves the way for targeting steroidogenesis and its signalling pathways in breast cancer affected by immune-steroid maladaptation.

Breast cancer remains a significant challenge in global health, ranking as the second most lethal cancer for women[1]. In 2020 alone, it accounted for over 2.3 million new diagnoses and caused 685,000 deaths, asserting its status as the most frequently diagnosed cancer worldwide[2]. Among its subtypes, triple-negative breast cancer (TNBC) emerges as a particularly aggressive type, and lacks specific targeted and effective therapy[3–6]. While immunotherapies mark the advent of a transformative period in the treatment of cancer[7–9], their effectiveness in TNBC has not been optimal, as indicated by only modest successes in clinical trials[10,11]. This underscores an imperative need to enhance immunotherapeutic outcomes for TNBC. Frontier research is thus channelled to unravel the complexities of the immunosuppressive tumour microenvironment (TME)[12–14]. A greater understanding of the mechanisms governing the host immune response within the TNBC TME may help identify novel therapeutic strategies.

The immunosuppressive microenvironment helps breast cancer cells evade anti-tumour immune responses[15–17]. Research focused on TNBC has identified that tumour-associated macrophages (TAMs) are

[1]Department of Pathology, University of Cambridge, Cambridge CB2 1QP, UK. [2]Cancer Research UK Cambridge Centre and Department of Oncology, University of Cambridge, Cambridge CB2 0XZ, UK. [3]Breast Cancer Center, Shandong Cancer Hospital and Institute, Shandong First Medical University and Shandong Academy of Medical Sciences, Jinan, Shandong 250117, China. [4]Mass Spectrometry Core, Edinburgh Clinical Research Facility, Queens Medical Research Institute, University of Edinburgh, Edinburgh, UK. [5]The Precision Breast Cancer Institute, Addenbrookes Hospital, Department of Oncology, University of Cambridge, Cambridge CB2 0QQ, UK. [6]These authors contributed equally: Qiuchen Zhao, Jhuma Pramanik. ✉e-mail: qiu.pf@outlook.com; bm562@cam.ac.uk

one of the major drivers of tumour growth and metastasis, largely due to their secretion of immunoinhibitory cytokines[18–20]. Additionally, FOXP3[+] regulatory T cells (Tregs), hinder the functions of both CD4[+] and CD8[+] T cells[21] and are associated with an unfavourable prognosis in TNBC[22]. Despite the accumulation of these suppressive immune elements, an active immune response is often stifled within the TNBC TME. While dendritic cells (DCs) are naturally poised to activate T cells, given their robust MHC I/II expression and abundance of co-stimulatory molecules[23], they can become compromised, leading to immune tolerance in dormant TNBC cell niches[24]. Further uncovering the underlying factors and cellular mechanisms that induce immune cell dysfunctionality within the TNBC TME is of paramount importance.

Steroidogenesis (steroid biosynthesis) is a biosynthetic process by which cholesterol is converted to steroid hormones[25]. The biosynthesis of steroids starting from cholesterol is often termed de novo steroidogenesis[25]. Cytoplasmic cholesterol is imported into the mitochondria, where the rate-limiting enzyme CYP11A1 (also known as P450 side chain cleavage enzyme) converts cholesterol to pregnenolone. Pregnenolone is the first bioactive steroid of the steroidogenesis pathway, and the precursor of all other steroids[25,26]. The steroidogenesis pathway has been extensively studied in the adrenal gland, gonads, and placenta. De novo steroidogenesis by other tissues, known as extra-glandular steroidogenesis, in brain[25,27], skin[28], thymus[29], and adipose tissues[30], has also been reported. Steroid production because of immune response in the mucosal tissues, such as in the lung and intestine, has been shown to play a tolerogenic role in maintaining tissue homeostasis[31,32]. Our lab and others discovered that certain immune cells, notably T lymphocytes and macrophages, can autonomously produce steroids within the TME. This finding was corroborated using B16-F10 melanoma and MC38 colon carcinoma syngeneic mouse models[33–35]. Evidence from these studies indicate that the process of steroidogenesis by immune cells hampers the anti-tumour immune response, specifically by weakening the effector functions of CD8[+] T cells, and driving macrophages towards an immunosuppressive phenotype[33]. However, it remains uncertain whether this mechanism of immune evasion is also relevant for human cancers and various cancer types. Given previous indications that TNBC tumours frequently provoke suppressive immune reactions, our next step is to delve into the influence of steroid-producing immune cells within the TME.

Steroid hormones, recognised as immunoregulatory agents, influence various immune processes[36,37]. For example, glucocorticoids (GCs) attenuate pro-inflammatory macrophages by inhibiting key inflammatory gene regulators, such as NF-κB and AP-1[38–40], and enhance the survival of anti-inflammatory macrophages by augmenting A3AR activation, an antecedent to anti-apoptotic pathways[41]. Additionally, GCs skew DCs towards a tolerogenic phenotype[42,43]. Dexamethasone-treated immature DCs lose the ability to mature fully and therefore, fail to prime Th1 immune response[43]. The majority of breast cancers respond positively (steroid sensitive) to the sex steroids, estrogen and progesterone, because most breast cancer types are either ER and/or PR positive. Estrogen and/or progesterone signaling promote their survival and growth. TNBC do not respond to sex steroids, but they do respond to glucocorticoids. Glucocorticoid signaling in breast cancer has a complex role and can be paradoxical depending on context. Overall, they promote TNBC by various means[44]. Glucocorticoid receptor (GR) activation can stimulate proliferation, help cells escape apoptosis, decrease cell adhesion and stimulate cell motility, which can increase the risk of metastasis[44]. High GR expression in early-stage TNBC is associated with chemotherapy resistance and increased recurrence[45]. GR expression can be a prognostic biomarker for TNBC[46]. Therefore, GR antagonists, such as mifepristone, may be used in conjunction with chemotherapy to treat TNBC[47–51]. Yet, the potential of de novo steroidogenesis in the TME and GR signalling to accelerate TNBC progression by sculpting an immunosuppressive microenvironment remains an uncharted territory.

Employing an integrative multi-omics approach, complemented by in vitro assays and in vivo experiments in mice models, we show the functional presence of steroidogenic immune cells within the TNBC TME, both in human and mice, and elucidate their role in breast cancer progression. Through steroid profiling and metabolic modelling at single-cell level, we delineate the specific steroids and discern their predominant producers within the TNBC TME. Inhibition of immune cell steroidogenesis by genetic deletion of the first and rate-limiting enzyme of the steroidogenesis pathway, Cyp11a1, in all immune cells, using *Cyp11a1*[fl/fl];*Vav1*[Cre] mice, restricts TNBC tumour progression by reinstating anti-tumour immune responses in dendritic cells and T cells by impeding glucocorticoid signalling. Inhibition of steroidogenesis pharmacologically using posaconazole, a drug that we identified by a structure-based in silico drug repurposing study, inhibits tumour growth in a humanised mouse model for TNBC. Overall, the study advances our understanding of the intricate immune-steroid interplay within the TNBC TME and paves the new way for therapeutic targeting.

## Results

### Functional Steroidogenesis and Steroid-Signalling Exist in TNBC Tumours

To detect the presence of steroidogenesis and steroid signalling within the tumour microenvironment (TME) of TNBC, we analysed tumour samples from 16 untreated primary TNBC patients (Fig. 1A; Supplementary Data 1). A targeted liquid chromatography tandem mass-spectrometry (LC-MS/MS) approach revealed 31 distinct steroid hormones, alongside 9 sterols, within these TNBC tumours (Fig. 1B; Supplementary Fig. 1A). Notably, dehydroepiandrosterone (an androgen) and cortisol (a glucocorticoid, GC) were the most predominant steroids present (Fig. 1B). We next analysed 360 primary TNBC patients' RNA-seq dataset[52]. The analysis revealed the glucocorticoid receptor *NR3C1* as the highly expressed steroid hormone receptor gene in TNBC tumours (Fig. 1C). To discern which steroid hormone dominates and exerts specific downstream signalling in the TME, we integrated metabolomics data on steroid hormone concentrations with transcriptomic data on genes and receptors related to steroids. This integration generated an overall score for each steroid in TNBC tumours, revealing corticosteroid hormones, especially cortisol, as having a superior score relative to other steroids (Fig. 1D; Supplementary Fig. 1B). This underscored the potential of GCs in modulating the TNBC TME. Further gene expression analyses revealed an inverse correlation between the expression of receptors for GC, androgen, and progestogen with the presence of activated DCs and M1 macrophages (Fig. 1E). By contrast, a positive correlation with resting mast cells, DCs, and M2 macrophages was observed, alluding to the potential inhibitory role of these steroids on myeloid populations in the TNBC TME (Fig. 1E).

Given the abundance of steroids and their predicted role in the TME, our next objective was to identify and delineate the de novo steroidogenic regions within tumour tissues. Haematoxylin and Eosin (H&E) and immunohistochemical (IHC) staining mapped elevated CYP11A1 expression - controlling the rate-limiting step of steroid biosynthesis[25]- within immune-infiltrated areas of the tumours (Fig. 1F). Semi-quantitative analysis based on IHC slides corroborated the pronounced steroidogenic propensity of tumour-infiltrating immune cells within TNBC tumours (Fig. 1G). Immunofluorescence (IF) staining also demonstrated co-localisation of CYP11A1 and CD45 (Fig. 1H). Further, a single cell RNA-seq (scRNA-seq) data assessment of 22 TNBC patients[53] showcased an enriched expression of genes pivotal for steroid biosynthesis within tumour-derived immune cells as compared to those from matched blood samples, establishing a robust local immune

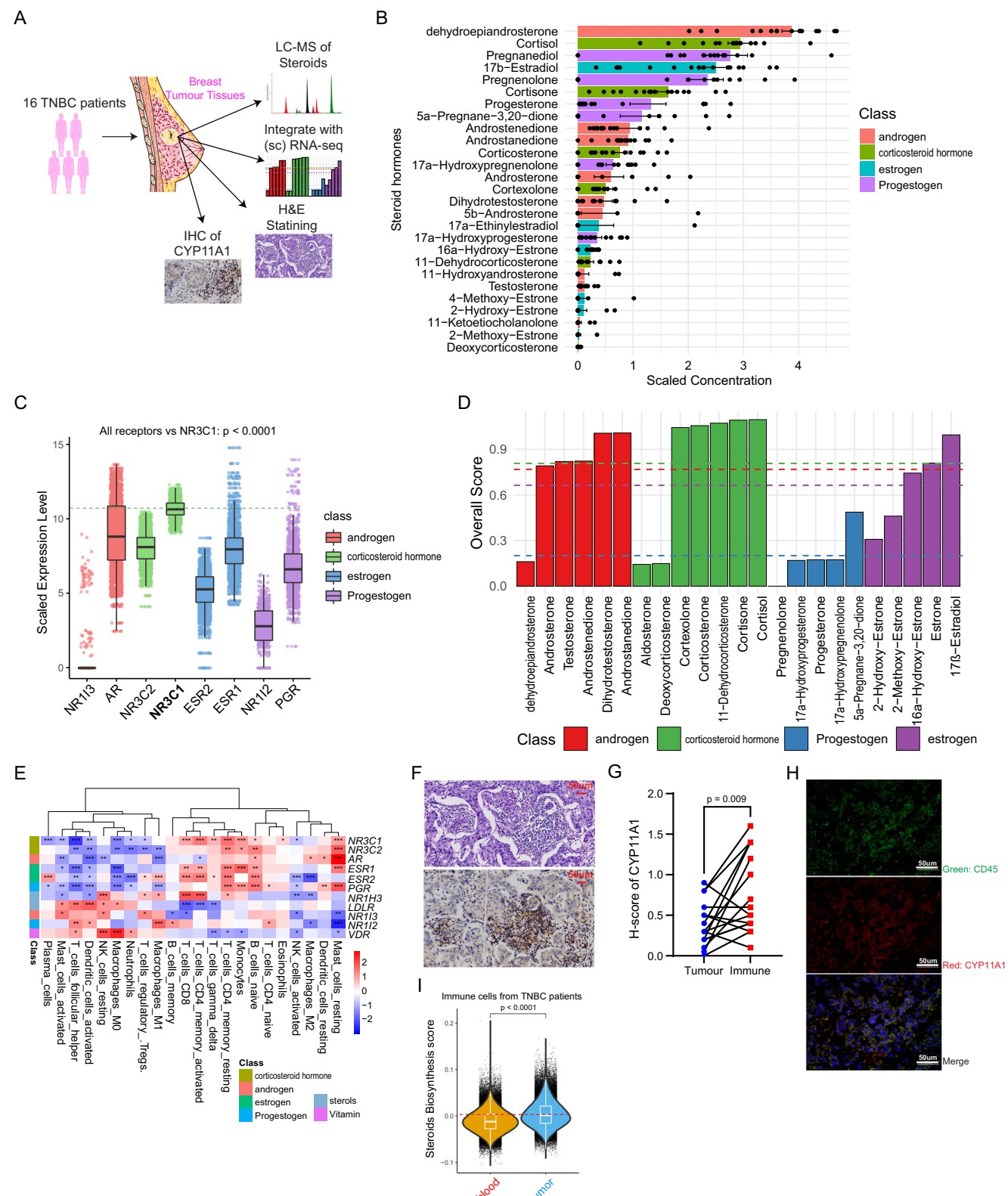

steroidogenesis signature in the TME (Fig. 1I; Supplementary Fig. 1C). A comparative analysis between *NR3C1*⁺ and *NR3C1*⁻ myeloid cells in the TME unveiled a pro-tumourigenic phenotype in *NR3C1*⁺ macrophages, marked by an upregulated expression of M2 macrophage-associated genes (Supplementary Fig. 1D). Concurrently, genes that are associated with antigen processing and presentation were diminished in *NR3C1*⁺ DCs in TNBC tumours (Supplementary Fig. 1E). Collectively, these findings confirm the presence of local steroidogenesis and

steroid-signalling in TNBC tumours and underscore the pivotal role of immune cells in steroid hormone production.

As expected, steroid hormone receptors, including glucocorticoids receptor, *NR3C1*, express in non-TNBC breast cancer (Supplementary Fig. 1F) in contrast to TNBC where glucocorticoids receptor is predominant (Fig. 1C). Along with these above findings, the absence of *CYP11A1* expression in TNBC cancer cells (Supplementary Fig. 1G) led us to hypothesise that immune cell steroidogenesis leads to local

**Fig. 1 | Local steroidogenesis and steroid signalling in TNBC tumours.**
**A** Schematic representation showcasing the integration of experimental approaches using 16 TNBC patient tumours. **B** Bar chart highlighting the scaled concentrations of 31 distinct steroid hormones detected in the 16 TNBC tumours via LC-MS (Mean ± SEM expressed through error bars); Source data are provided as a Source Data file. **C** Box plot detailing the distribution of the most abundant steroid hormone receptor, *NR3C1*, across TNBC tumours, derived from RNA-seq data (Jiang. et al.) from a cohort of 360 TNBC patients (NR1I3: Nuclear Receptor subfamily 1 group I member 3, AR: Androgen Receptor; NR3C1 & NR3C2: Nuclear Receptor Subfamily 3 Group C Member 1 & 2, ESR1 and 2: Estrogen Receptor 1 and 2; PGR: Progesterone Receptor); **D** Bar chart indicating the overall steroid hormone score derived by integrating steroid hormone concentrations and gene expression data in TNBC tumours. **E** Heatmap displaying the correlation between steroid receptor expression levels and different immune populations, derived from RNA-seq data (Jiang. et al.) from a cohort of 360 TNBC patients (a unpaired two-tailed t-test was performed). **F** Hematoxylin and Eosin (H&E) staining (top) and Immunohistochemistry (IHC) staining for CYP11A1 (bottom) in TNBC tumours, showcasing differential localization patterns. Image represents data from one representative of 16 TNBC patients. **G** Visual representation indicating the CYP11A1 activity levels across different regions of the 16 TNBC tumours (a unpaired two-tailed t-test was performed); Source data are provided as a Source Data file. **H** Representative immunofluorescence images of tumour sections of one out of four TNBC patients showing CD45 (green), CYP11A1 (red), and overlap. **I** Violin plot detailing the comparative levels of steroid biosynthesis scores in immune cells derived from tumour samples versus blood samples, based on scRNA-seq data from 22 TNBC patients. In (**C**) and (**I**), the median is used as the center, the 25th and 75th percentiles are set as the boundaries, whiskers are drawn to 1.5×IQR limits, and outliers are plotted separately; a unpaired two-tailed t-test was performed. Figure 1A was partly generated using Servier Medical Art, licensed under a CC BY 4.0 license.

production of immunosuppressive steroids (e.g., glucocorticoids) that suppress anti-tumour immunity by hampering immune cell function.

## Genetic Deletion of Cyp11a1 in Immune Cells Restricts TNBC Tumour Growth and Alters Immune Infiltration in the TME

We hypothesised that inhibiting immune cell-mediated steroidogenesis would enhance immune-mediated anti-tumour responses. To test this hypothesis, we generated a pan-hematopoietic cell-specific *Cyp11a1* null mouse model *Cyp11a1*$^{fl/fl}$;*Vav1*$^{Cre}$ (Cyp11a1$^{cKO}$). We chose pan-haematopoietic *Cyp11a1* null mice because previous studies indicated that induction of immune cell steroidogenesis is context-dependent and multiple immune cell types (e.g., T cells, macrophages, mast cells and basophils) might be involved[33,54,55]. This conditional knockout precluded steroidogenesis across all immune cell types due to the absence of Cyp11a1 (Fig. 2A, Supplementary Fig. 2A). E0771.LMB cells were implanted subcutaneously into these mice (Fig. 2A). The Cyp11a1$^{cKO}$ mice displayed a pronounced decline in tumour growth rates when contrasted with control mice (Fig. 2B, C) possibly because intratumoural steroidogenesis was abolished (Fig. 2D). Cancer cells (E0771.LMB) cells do not produce steroids (Supplementary Fig. 2B). The Cyp11a1$^{cKO}$ mice show no significant difference in the systemic level of steroid hormones as revealed by steroid profiling of serum samples (Supplementary Fig. 2C).

To uncover the cellular and transcriptional variations within the tumour-infiltrating immune cells, scRNA-seq was performed on CD45$^+$ immune cells isolated from TNBC tumours using fluorescence-activated cell sorting (FACS) (Supplementary Fig. 2D). This comprehensive profiling unveiled prominent shifts in the majority of immune cell types when comparing control (*Vav1*$^{Cre}$) and Cyp11a1$^{cKO}$ mice (Fig. 2E, F & Supplementary Fig. 2E). The relative percentage of each principal immune cell type varied significantly between control and Cyp11a1$^{cKO}$ specimens. The proportion of B cells, basophils, *Ccr7*$^+$ DCs, and macrophages diminished, while there was an increase in *cKit*$^+$ NK and traditional NK cells in the Cyp11a1$^{cKO}$ cohort when compared against the control group (Fig. 2G). A notable finding was the marked reduction of macrophages in the Cyp11a1$^{cKO}$ samples (Fig. 2G), with their genetic expression leaning more towards an anti-tumour M1 type (Fig. 2H & I). Flow cytometry analyses further strengthened these observations, spotlighting a reduction in the tumour-favouring Mertk$^+$ and Arg1 + M2 macrophages (Fig. 2J & L), while showing an increase in the iNos$^+$ M1 phenotype (Fig. 2K). Although there were no dramatic changes in the overall T cell population, subtle distinct variations in their subsets were observed when comparing the two groups (Supplementary Fig. 2F & G). In the Cyp11a1$^{cKO}$ specimens, there was a noticeable decline in *Foxp3*$^+$ Tregs and *Pdcd1*(encodes PD-1)$^+$ CD8$^+$ T cells (Supplementary Fig. 2H & I). However, no significant differences were found in the relative percentages of these immune populations or in the expression of their functional markers in spleen samples from

Cyp11a1$^{cKO}$ and control mice (Supplementary Fig. 2J–Q). All in all, our findings show that the genetic targeting of Cyp11a1 in immune cells curtails the growth of TNBC tumours and impinges upon the infiltration of immunosuppressive cells within the TME.

## Steroidogenesis Inhibition Augments Anti-Tumour Immunity by Suppressing Glucocorticoid Signalling

We next sought to define specific steroid signalling pathway that could be revitalising anti-tumour immunity. Our comprehensive scRNA-seq analysis of tumour-infiltrating immune cells from TNBC patients highlighted *NR3C1* as the dominant steroid receptor gene (Fig. 3A), a finding that is consistent with data from our TNBC mouse model (Supplementary Fig. 3A). This receptor's significance was emphasised by the reduced expression of the glucocorticoid-inducible factor *Tsc22d3* (also known as glucocorticoid induced leucine zipper or *GILZ*) in Cyp11a1$^{cKO}$ tumour infiltrated immune cells, compared to control tumours (Fig. 3B). Notably, *Tsc22d3* emerged as the most significantly downregulated gene in DCs from Cyp11a1$^{cKO}$ mice, and the expression levels of *Ddit4* and *Fkbp5*, which are known to regulate glucocorticoid responses and immunosuppression, were also diminished in DCs from Cyp11a1$^{cKO}$ mice. By contrast, genes crucial for dendritic cell function like *Cd209e* and *Cd44* were upregulated (Fig. 3C).

To further delineate the role of GCs in immune suppression, we exposed human DCs with cortisol, the functional human endogenous glucocorticoid, and then subjected these to RNA-seq, flow cytometry, and ELISA (Fig. 3D). The impact of cortisol was evident on the transcriptional profile of DCs (Supplementary Fig. 3B), inducing up-regulation of genes such as *IL1R2* and *FKBP5* while depressing others like *IL12A* (Supplementary Fig. 3C, Supplementary Data 2). Notably, GC-exposed DCs exhibited diminished co-stimulatory factors like *CD86* and immune-active cytokines like *IL12A* and *IL6* (Fig. 3E). In contrast, there was an increase in immunoregulatory cytokines, such as *TGFB1* and *IL10* (Fig. 3E), a pattern validated via flow cytometry and ELISA (Fig. 3F, G). Gene set enrichment analysis (GSEA) further revealed that DCs exhibited a stronger response to GC, a more tolerogenic phenotype, and decreased gene activity related to regulation of T cell cytotoxicity and CD4$^+$ T cell activation (Supplementary Fig. 3D–G). Subsequent mixed lymphocyte reaction (MLR) assays, pairing GC-treated and mock DCs with naïve T cells, identified a correlating decline in IFNγ$^+$ and TNFα$^+$ cytotoxic CD8$^+$ T cells (Fig. 3H & I).

Importantly, steroidogenesis inhibition by deleting *Cyp11a1* (Cyp11a1$^{cKO}$ mice) effectively countered these GC-induced alterations, as revealed by up-regulated genes from RNA-seq data, in tumour infiltrating DCs of Cyp11a1$^{cKO}$ tumour-bearing mice compared to control tumour bearing mice (Fig. 3J). And the dendritic cells presented less exhaustion and displayed a mature phenotype (Supplementary Fig. 3H & I). IL-10 and TGF-β were also reduced in DCs from Cyp11a1$^{cKO}$ tumours (Fig. 3K & Supplementary Fig. 3J). In

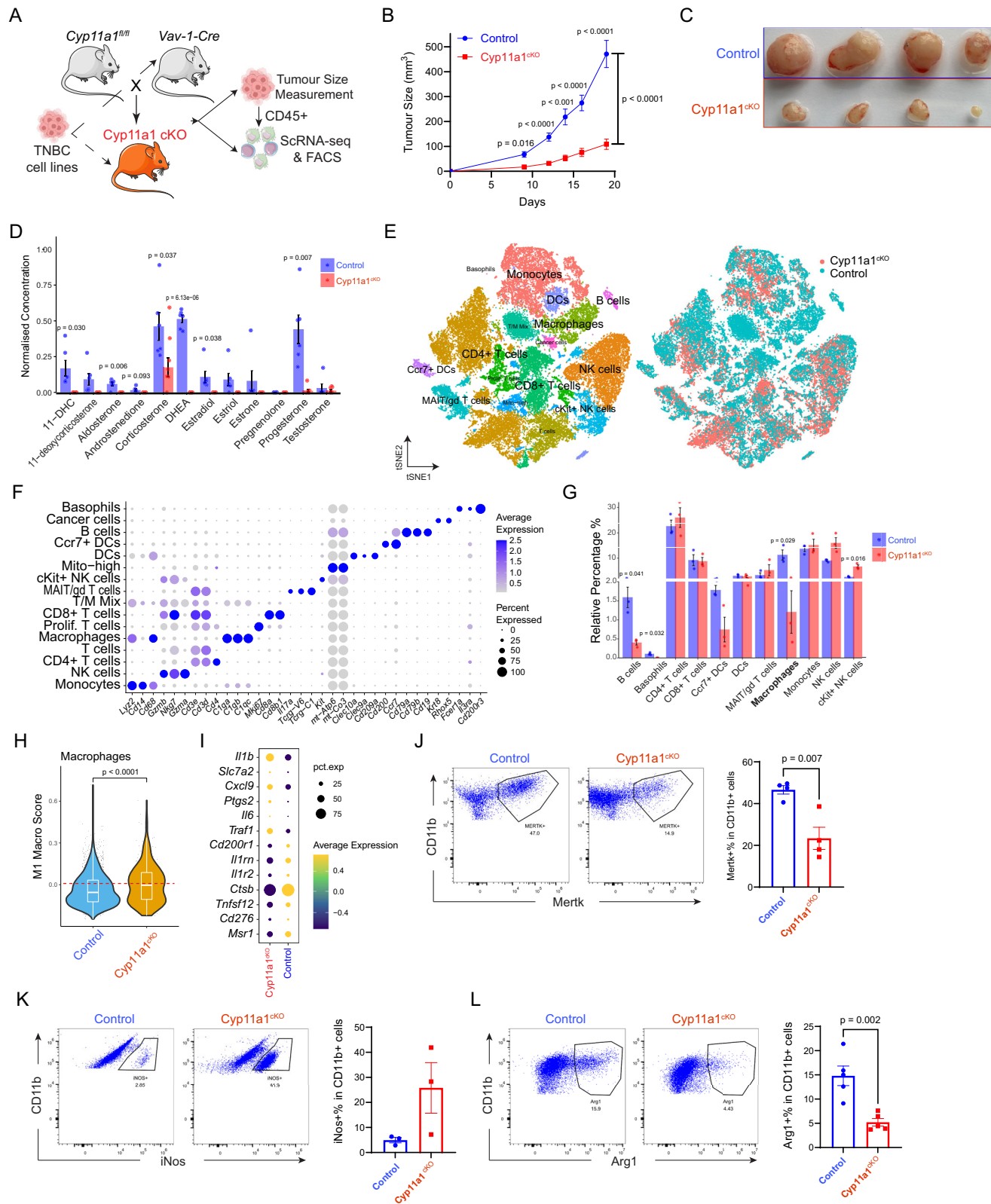

parallel, Cyp11a1$^{cKO}$ tumours housed CD8$^+$ T cells with heightened activation markers and decreased exhaustion traits (Fig. 3L & Supplementary Fig. 3K). This was further confirmed by elevated IFNγ+ and TNFα+ cytotoxic CD8$^+$ T cells within the Cyp11a1$^{cKO}$ tumour milieu (Fig. 3M & N). Collectively, our findings underscore that inhibiting steroidogenesis disrupts glucocorticoid immunosuppression, bolstering anti-tumour immunity.

## Characterisation of Cyp11a1$^+$ Immunocytes in TNBC for Enhanced Therapeutic Precision

We next aim to identify the prominent immune cell subsets responsible for initiating the steroidogenesis cascade and ensuring their secretion of immunosuppressive steroid hormones in the TME. Following the implantation of the E0771.LMB cells into Cyp11a1-mCherry reporter mice, we profiled the steroid hormones by LC-MS/MS, and

**Fig. 2 | Genetic deletion of Cyp11a1 in immune cells alters TNBC tumour dynamics and the immune landscape of the TME. A** Schematic illustration depicting the experimental strategy. **B** Subcutaneous E0771.LMB tumour growth curve of control versus Cyp11a1$^{cKO}$ mice; each data point represents average tumour size (N = 12; Two-way ANOVA for growth curves). **C** Representative photograph of E0771.LMB tumour in control and Cyp11a1$^{cKO}$ mice. **D** Bar chart showing the scaled concentrations of steroid hormones detected in the tumours from control and Cyp11a1$^{cKO}$ mice via LC-MS (N = 6 for control and N = 9 for Cyp11a1$^{cKO}$). **E** t-SNE plots of CD45$^+$ immune cells from control (N = 3) and Cyp11a1$^{cKO}$ (N = 3) mice. Different colours on the left highlight distinct immune cell subsets, while the right contrasts cell origins from control versus Cyp11a1$^{cKO}$ mice. **F** Marker gene expression across immune cell clusters of panel (**E**). **G** Relative percentages of immune cell populations between Vav1$^{Cre}$ control and Cyp11a1$^{cKO}$ mice (N = 3). **H** Differential M1 macrophage scores in macrophages extracted from Vav1$^{Cre}$ control versus Cyp11a1$^{cKO}$

mice. Each dot represents a single cell. The box plot shows the median as the center, the 25th and 75th percentiles as the box boundaries, whiskers extending to the most extreme values within 1.5×IQR from the quartiles, and outliers plotted separately. **I** Dot plot presentation of M1 versus M2 gene expression in macrophages sourced from both Vav1$^{Cre}$ control and Cyp11a1$^{cKO}$ mice. **J–L** Flow cytometry profiles emphasising Mertk (**J**), iNos (**K**) and Arg1 (**L**) expression in CD11b$^+$ cells from Vav1$^{Cre}$ and Cyp11a1$^{cKO}$ mice (left). The adjoining chart (right) showcases the percentage expression for each group. N = 4 (**J**), N = 3 (**K**), N = 5 (**L**). All experiments are representative of two or three independent repeats except (**D–G**). All error bars except H: Mean ± SEM; p values were calculated using unpaired two-tailed t-test unless stated. Source data for **B**, **D**, **J** & **K** are provided as a Source Data file. Figure 2A was partly generated using Servier Medical Art, licensed under a CC BY 4.0 license.

isolated CD45$^+$mCherry$^+$ and CD45$^-$mCherry$^-$ tumour-residing immune cells for in-depth scRNA-seq and flow cytometric assessments (Fig. 4A). Our scRNA-seq analysis delineated dominant immune categories within the mCherry$^+$ and mCherry$^-$ groups, annotated based on canonical marker gene expression (Fig. 4B & C). Notably, the Cyp11a1 expressing (i.e., mCherry$^+$) steroidogenic immune cells predominantly encompassed macrophages, mast cells, and basophils, with residual populations comprising monocytes, T cells, B cells, and NK cells (Fig. 4B & Supplementary Fig. 4A).

To reveal metabolic trajectories at the cellular level, we employed single-cell Flux Balance Analysis (scFBA)[56]. Recognising the essential function of CYP11A1 in transitioning cholesterol to pregnenolone, mast cells and basophils demonstrated higher metabolic flux compared to various macrophage clusters (Fig. 4D). Subsequent investigations into downstream steroid hormones and their respective synthesisers within TNBC tumours revealed elevated fluxes in mast cells and basophils, leading to increased synthesis of steroids, notably corticosterone and DHEA (Fig. 4E).

Our observations corroborated the scFBA findings from TNBC patient samples highlighting mast cells' role in steroidogenesis, but human datasets lacked basophil representation (Fig. 4F). In agreement with these results, we observed a negative association surfaced between mast cell presence together with CYP11A1 expression and the survival trajectories of breast cancer patients (Fig. 4G). As expected, FceR1$^+$cKit$^+$SiglecF$^-$ mast cells in tumours from the Cyp11a1-mCherry reporter mice distinctly displayed mCherry signals (Supplementary Fig. 4B). Mast cells expressing Cyp11a1 also manifested elevated mast cell activation gene expression (Supplementary Fig. 4C).

Having identified the TNBC TME's steroidogenesis initiator cells, we tested posaconazole, a Cyp11a1 inhibitor revealed by a drug repurposing study[57], on Cyp11a1 activity in peritoneal FceR1$^+$cKit$^+$ mast cells (Fig. 4H & 4I). Treatment with posaconazole resulted in a significant reduction in pregnenolone levels, without inducing notable cell mortality (Fig. 4J, Supplementary Fig. 4D).

**Posaconazole Diminishes TNBC Progression in Preclinical Models**

Following the elucidation of posaconazole's capacity to impede de novo steroidogenesis by inhibiting Cyp11a1, we explored its therapeutic potential against CYP11A1 in vivo (Fig. 5A) in TNBC model (i.e., E0771.LMB subcutaneous syngeneic model). Remarkably, posaconazole-treated mice displayed a significant restriction in TNBC tumour progression relative to their control counterparts (Fig. 5B). This inhibition was further corroborated by the reduced end-point tumour sizes (Fig. 5C). Genome-wide transcriptomic changes (RNAseq) in posaconazole-treated tumour samples show an improved effector immune activity both in the innate and adaptive immune cells compared to untreated samples (Fig. 5D). Immunophenotyping by flow cytometry revealed a surge in cytotoxic CD8$^+$ T cells in posaconazole-

treated mice, which were marked by heightened IFNγ and TNFα expression (Fig. 5E, F).

To validate posaconazole's potential for clinical translation, we employed the immune-humanised huPBMC-NOG-dKO (murine MHC class I- and class II-deficient) mouse model, and human TNBC cells were orthotopically introduced into the breast fat pad (Fig. 5G). Subsequent magnetic resonance imaging (MRI) scans, performed after a week, unveiled a pronounced tumour growth restriction in drug-administered mice (Fig. 5H). Three-dimensional models inferred from MRI data reinforced the trend of posaconazole curtailing tumour growth (Fig. 5I). The drug was well-tolerated with no adverse effects noted, and it maintained an improved body weight in drug-treated mice, in contrast to weight loss in the control group (Supplementary Fig. 4E). Tumours harvested post 21 days from these humanised mice further underlined posaconazole's inhibitory prowess (Fig. 5J & K). GSEA based on RNA-seq further revealed that a robust *IFNG* and *TNFA* responses within the TNBC tumour microenvironment following posaconazole treatment (Fig. 5L & M). Our findings underscore the potential of posaconazole to stall TNBC progression in preclinical settings, suggesting its therapeutic promise for breast cancer.

## Discussion

In our quest to discover modes of interplay between TNBC and the immune system, we have shown here that immune cell steroidogenesis suppresses anti-tumour immunity, that can be reinstated by targeting the pathway. Profiling of steroids within TNBC tumours illuminates an overlooked aspect of tumour immunology, revealing a robust association between heightened immune cell steroidogenesis and a dampened anti-tumour response in an autocrine and paracrine manner. Local steroid biosynthesis and steroid-signalling induce tolerance in DC, increased FoxP3+ T regulatory cells, increased M2:M1 ratio and dysregulated CD8 T cells. Tumour restriction because of *Cyp11a1* deletion underscore the therapeutic potential of targeting steroidogenesis, by stimulating immune responses against TNBC (Fig. 6). At the cellular level, in TNBC, mast cells, basophils and macrophages emerged as the key initiator of steroidogenesis, and all major effector immune cell types appeared as responder to local steroid, pointing to potential targets for treatment. Moreover, our discovery of posaconazole as an effective pharmacological inhibitor supports our genetic interventions and opens new possibilities for therapeutic advancements in TNBC by repurposing this drug. This research elucidates the profound implications of immune-steroid signal transduction dynamics, suggesting a promising treatment for TNBC in the future.

The relationship between steroid hormones and the TME in TNBC has been elusive due to previous untargeted metabolomics approaches, which lacked precision, and most of the early TNBC patients had neoadjuvant therapy before surgery, possibly changing local hormone levels[58–60]. Our approach of analysing tumour samples from the patients that have not gone through therapy yet, and use of more customised extraction methods for precise LC-MS techniques, has

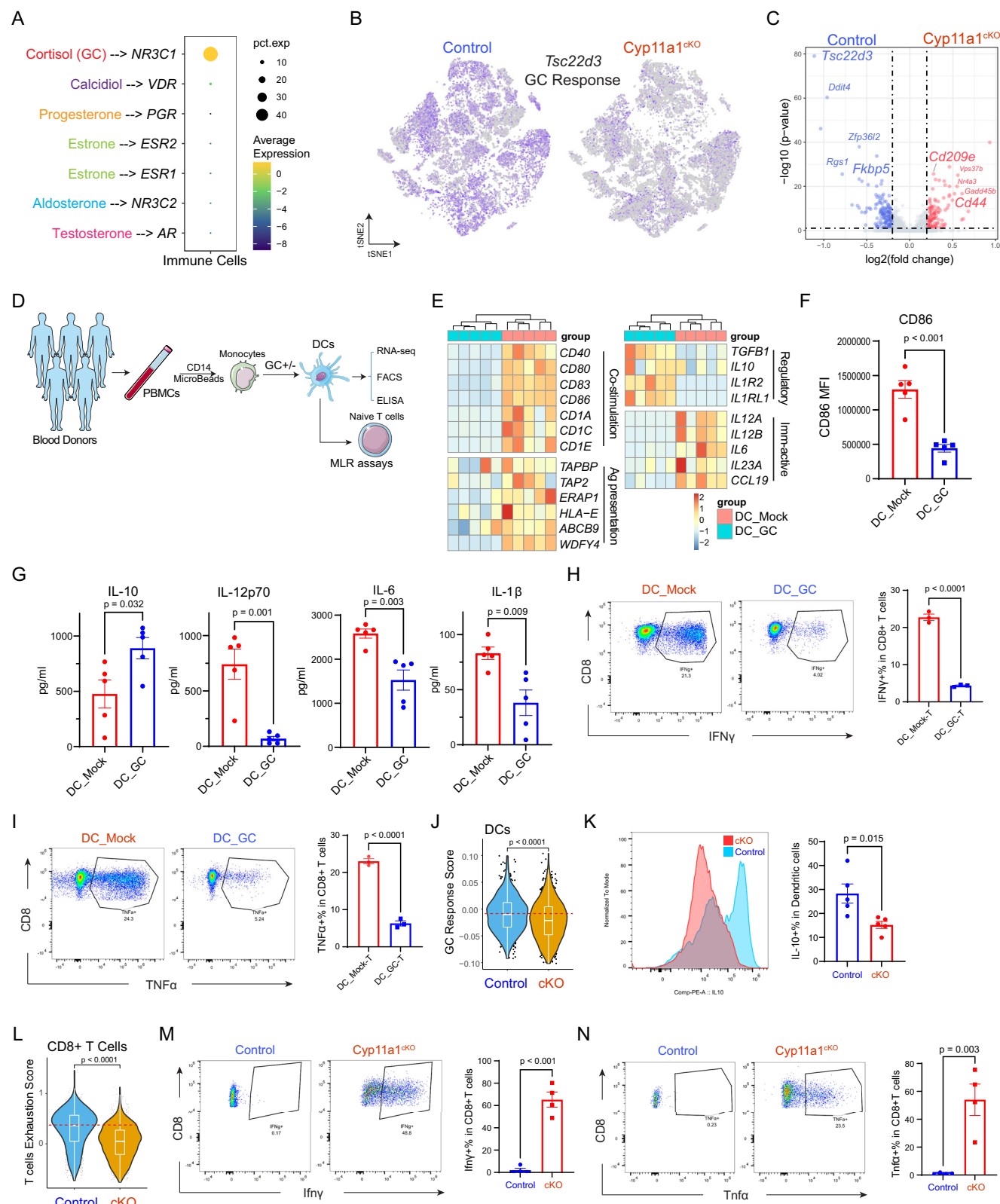

revealed a variety of steroid hormones in tumours from untreated TNBC patients, highlighting the unexpected prominence of glucocorticoids, particularly cortisol (Fig. 1B). This finding, supported by evident *NR3C1* expression in TNBC tumours (Fig. 1C), adds new perspectives to our understanding of TNBC. By integrating steroidomics and transcriptomics, we gain a deeper understanding of how certain steroid hormones may impact signalling in the TME (Fig. 1D). This

innovative approach has potential applications in biology for studying steroid signalling with multi-omics. We also discovered that tumour-infiltrating immune cells are significantly geared towards steroid production, evident from high levels of CYP11A1 expression and existence of a steroidogenic-steroid-responsive gene signature in the patients' single-cell RNA sequencing data (Fig. 1), supporting the hypothesis that steroids play a crucial role in immune modulation within the TME. In

**Fig. 3 | Steroidogenesis inhibition augments anti-tumour immunity by suppressing glucocorticoid (GC) signalling. A** Dot plot delineate the prevalence of steroid hormone receptor genes in tumour-infiltrating immune cells from TNBC patients. **B** Comparison of *Tsc22d3* expression within immune cells of Vav1^Cre and Cyp11a1^cKO mice (sc-RNAseq). **C** Volcano plot depicting the differential gene expression in DCs post-steroidogenesis inhibition. *p* values and fold changes calculated using Seurat (version 3.2.2.). **D** Experimental design for **E–I** Human PBMC-derived monocytes were differentiated toward DCs +/− GC. **E** Expression levels of selected genes between GC-treated (DC-Mock) and treated (DC-GC) groups. **F** CD86 expression (FACS) in DC-Mock and DC-GC groups (N = 5). **G** Cytokine concentrations (ELISA) in DC-Mock and DC-GC groups (N = 5). **H–I** Representative FACS profiles (left panel) and average percentage expression (right panel) of IFNg (**H**) and TNFa (**I**) in CD8 ⁺ T cells from the mixed lymphocyte response assay (naïve T and DC co-culture assay, T-DC-Mock and T-DC-GC groups, N = 3, biological replicates. **J** Violin plots revealing indicated gene set signature scores in DCs from

scRNA-seq data of tumours of either Vav1^Cre or Cyp11a1^cKO mice. **K** Representative FACS histogram (left) and bar graph (right) displaying the expression intensity and percentage of IL-10 in DCs from Vav1^Cre or Cyp11a1^cKO mice (N = 5). **L** Violin plots reveal the indicated gene set signature scores in CD8 + T cells from scRNA-seq data of tumours of Vav1^Cre and Cyp11a1^cKO mice. **M–N** FACS diagrams (left), and bar graph (right) displaying the expression intensity and percentage of Ifng (**L**) and Tnfa (**M**) in CD8 ⁺ T cells from Vav1^Cre or Cyp11a1^cKO mice (N = 4). In all violin plots, each dot represents a single cell. The box plot shows the median as the center, the 25th and 75th percentiles as the box boundaries, whiskers extending to the most extreme values within 1.5×IQR from the quartiles, and outliers plotted separately. All error bars are Mean ± SEM. All *p* values were calculated by unpaired two-tailed t-test unless stated. Source data for **F–K**, **M**, **N** are provided as a Source Data file. **N** defines each individual donor or mouse or independent biological replicates. Figure 3D was partly generated using Servier Medical Art, licensed under a CC BY 4.0 license.

the future, CYP11A1 IHC may help diagnose the steroidogenic tumours to stratify patients. It would be valuable to segregate clinically AR-positive and AR-negative TNBC samples to analyse and compare their overall scores, potentially confirming higher corticosteroid levels and further elucidating the relationship between AR status and steroid hormone profiles in TNBC subtypes.

By eliminating CYP11A1-mediated steroidogenesis in immune cells, we ventured to understand its influence on TNBC tumour progression. The marked decline in tumour growth in the Cyp11a1^cKO mice underscored CYP11A1's significance (Fig. 2C) and promises for its use as a drug target.

Modern advances in single-cell technologies offer in-depth characterisations of tumour-infiltrating immune cells[61,62], and our use of single-cell transcriptomics approach revealed the profound influence of CYP11A1 on immune cell infiltration (Fig. 2F). A striking finding was the intricate relationship between CYP11A1 and GC signalling in the TNBC, which was also explored by single-cell transcriptomics approach (Fig. 3B). Previous studies have indicated the GC-mediated immunosuppression in dendritic cells[63]. Our data illustrated the correlation between GC signalling and dendritic cell function, particularly the pronounced changes in gene expression and cytokine profiles (Fig. 3D–I), underscores the influence of steroids on the orchestration of anti-tumour immunity.

Observation of Cyp11a1-mCherry⁺ myeloid cells (macrophages/ monocytes) in the tumour-bearing Cyp11a1-mCherry⁺ mice was consistent with the previous observation of colorectal cancer by Acharya et al., Immunity, 2020[54]; but dissimilar with melanoma by Mahata et al., Nat Commun, 2020[33], where the percentage of Cyp11a1⁺ macrophages/monocytes were very low by number. This observation indicates the heterogeneity of the tumour microenvironment. Induction of immune cell steroidogenesis in immune cell types seems to be tumour-type and mice model-specific. Observation of mast cell expression of Cyp11a1 in breast cancer never been reported. Mast cells, traditionally perceived as key players in allergic responses and inflammatory processes, have in recent years, gained recognition for their multifaceted roles within the tumour development. Previous reports claim increased mast cell presence with enhanced breast cancer angiogenesis[50,51]. Our study suggests that mast cells modulate immune responses in the TNBC TME by contributing to the local immunosuppressive steroid biosynthesis (Fig. 4D, F). The immune suppressive capacity of mast cells is attributed by the secretion of IL-10, TGFβ, prostaglandin D2's that skew dendritic cell function driving a Th2 immune response[64,65]. Our identification of Cyp11a1⁺ steroidogenic mast cells within the TNBC TME posits them as instrumental in producing immunomodulatory steroids, aiding the immune escape tactics of cancer cells, possibly by inducing tolerance in DCs. However, further experimentation is needed for direct evidence. Functionally similar basophil's role in this study is not clear, particularly in humans.

The path from laboratory discovery to clinical application is intricate and challenging. Nevertheless, a drug repurposing study (Pramanik and Shaji et al., 2025)[57] has pointed us towards posaconazole. This drug has a known pharmacokinetic and pharmacodynamic profile, which promises a quicker transition from the lab bench to the clinic[66]. Posaconazole, a well-established antifungal medication used worldwide, exerts its effects through the inhibition of lanosterol 14α-demethylase, an enzyme vital for ergosterol synthesis, which is crucial for maintaining fungal cell membrane integrity[67]. Interestingly, posaconazole has been proposed as potential drug for basal cell carcinoma and glioblastoma[68,69], and improved survival in acute myeloid leukaemia cases with fungal infections[70]. Importantly, our data strongly suggest that posaconazole could be effective in treating TNBC, showcasing a new potential mechanism for stimulating anti-tumour immunity. The huPBMC-NOG-dKO mouse model provides substantial support for this, elevating posaconazole to a prime position for clinical testing. Clear signs of tumour reduction in both MRI scans and measurements of final tumours (Fig. 5G, I) reinforce its potential. Moreover, the treatment showed a favourable safety profile, with no noted adverse effects including body weight in humanised mice model. Moving forward, transitioning these positive preclinical results into clinical trials will be essential. Despite the need for comprehensive and stringent testing to unveil posaconazole's full potential and identify any potential risks with combination therapies, the current evidence positions it as a promising agent for advancing TNBC therapy.

Steroid biosynthesis mainly occurs in steroidogenic glands such as the adrenal gland, ovary, testis, and placenta. Extra-glandular or local steroid biosynthesis is also reported in multiple tissues in a context-dependent manner[40,71]. The possibility of maladaptation of local steroid synthesis and signalling has also been reported in cancer[40,71]. In colorectal cancer and melanoma, evidence is stronger[33–35,72]. These reports raised the possibility that maladapted steroid signalling may frequently happen in many cancer types at an unprecedented level. This study shows evidence that this prediction is true in breast cancer, particularly in TNBC both in human and mice, and the pathway can be targeted pharmacologically.

CYP11A1 is the first and rate-limiting enzyme of the steroidogenesis pathway[25]. However, recent studies suggest that it can activate vitamin D3, lumisterol and tachysterol to biologically active metabolites[73]. Therefore, local expression of this enzyme in TNBC could play a homeostatic role by producing these metabolites. Cancer not only escapes the body's regulatory mechanisms, but it also modulates local and systemic homeostasis, favouring tumour growth. Tumours produce cytokines, immune mediators, classical neurotransmitters, hypothalamic and pituitary hormones, many metabolites, and glucocorticoids to control neuroendocrine regulation of homeostasis as demonstrated in human and animal cancer models[74]. Cyp11a1 activity, therefore, can mediate bidirectional communication between local autonomic and sensory nerves, and the tumour, with

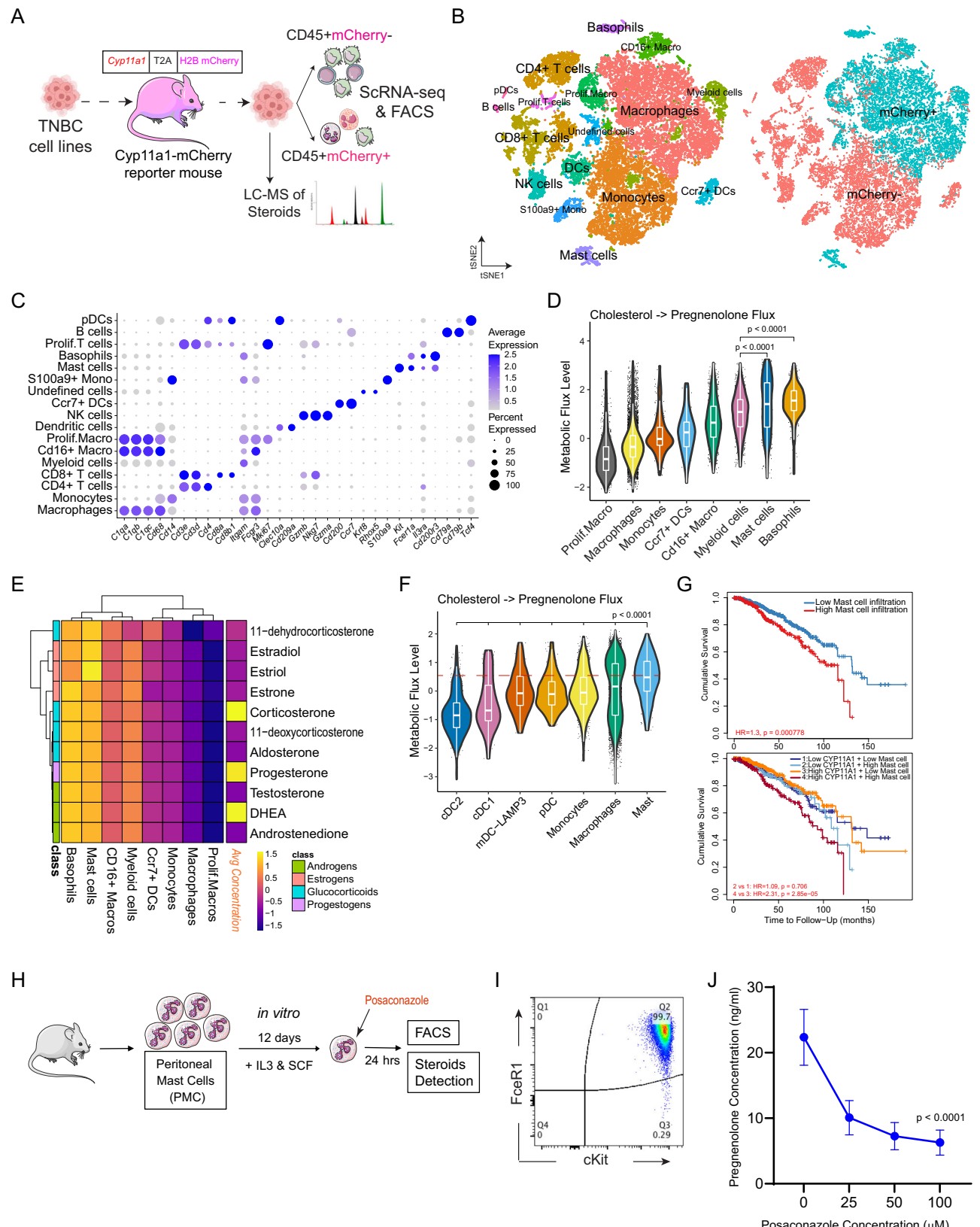

putative effects on the brain, is also envisioned. In this study we cannot exclude the possibility of posaconazole to inhibit such metabolite production beyond its ability inhibit steroidogenesis.

In summary, this study uncovers a previously unknown interactions between TNBC and the immune system, highlighting the promising therapeutic potential of targeting local steroidogenesis

(Fig. 6). The demonstrated effectiveness of posaconazole in pre-clinical models signals a potential shift in treatment strategies for TNBC. As we approach the clinical trial phase, we are optimistic that these findings will pave the way for the development of more precise and effective treatments for individuals affected by TNBC.

**Fig. 4 | Characterization of Cyp11a1⁺ immunocytes in TNBC for enhanced therapeutic precision. A** Outline of the Cyp11a1-mCherry reporter mouse experiment subcutaneously transplanted with E0771.LMB cells, showcasing tumour-infiltrating immune cell (CD45⁺mCherry⁺ and CD45⁺mCherry⁻ cells) sorting for scRNAseq and steroid hormone analysis (**B–E**). **B** t-SNE visualization of distinct clusters in CD45⁺mCherry⁺ and CD45⁺mCherry⁻ immune cells from the tumour bearing reporter mice. Different colours on the left highlight distinct immune cell subsets, while the right contrasts cell origins from CD45⁺mCherry⁺ and mCherry⁻ population. **C** Marker gene expression for various cell clusters displayed via a dot plot. **D** Violin plot representation of flux levels from cholesterol to pregnenolone within CD45⁺mCherry⁺ immune cells from tumour samples. **E** Detailed heatmap showing variations in flux levels of steroid hormone production among different immunocytes derived from CD45⁺mCherry⁺ cells, along with the concentrations of these steroids in tumour samples. **F** Violin plot representation depicting flux levels from cholesterol to pregnenolone within myeloid cells from 22 TNBC patients.

**G** Kaplan-Meier curve elucidating the correlation between both mast cell infiltration (up) and the combined impact of CYP11A1 mRNA expression with mast cell infiltration (bottom) on survival outcomes in breast cancer (BRCA) patients, analysed using the TIMER database (a two-sided p-value was derived from the standard normal distribution). **H** Schematic diagram of the in vitro Posaconazole targeting strategy on mast cells. **I** Flow cytometry scatter plots showcasing FceR1 and cKit expression in mast cells. **J** Data showcasing Posaconazole impact on the pregnenolone production of mast cells ($N = 4$ independent biological replicates; Mean ± SEM demarcated by error bars; one-way ANOVA was performed); Source data are provided as a Source Data file. In all violin plots (**D, F**), each dot represents a single cell. The box plot shows the median as the center, the 25th and 75th percentiles as the box boundaries, whiskers extending to the most extreme values within 1.5×IQR from the quartiles, and outliers plotted separately. $P$ values were calculated using unpaired two-tailed t-test unless stated. Figure 4A, H were partly generated using Servier Medical Art, licensed under a CC BY 4.0 license.

## Methods

### Human Specimens
In this study, 16 female patients diagnosed with early-stage triple-negative breast cancer (TNBC) were identified at Shandong Cancer Hospital and Institute (Jinan, China). Eligibility criteria included a histological confirmation of TNBC, no prior exposure to anti-tumour therapies, and surgical treatment. All research protocols were approved by the Ethical Committee of Shandong Cancer Hospital and Institute, affiliated with Shandong First Medical University and Shandong Academy of Medical Sciences (Approval No. SDTHEC2021003035). Every participant provided written informed consent, permitting tissue collection and subsequent metabolomics profiling. Regarding surgical details, 10 underwent mastectomy, 4 had lumpectomy, and 2 opted for reconstruction. Collected breast tumour specimens were immediately stored in liquid nitrogen. Detailed participant data is provided in Supplementary Data 1 of the supplementary materials.

### Mice
In this investigation, all mice were handled and cared for following the stringent guidelines set forth by the UK Animals in Science Regulation Unit's Code of Practice for the Housing and Care of Animals Bred, Supplied, or Used for Scientific Purposes, and the Animals (Scientific Procedures) Act 1986 Amendment Regulations 2012. All experimental protocols were conducted under the authorisation of a UK Home Office Project License (PPL P0AB4361E) and received the necessary approval from the local institute's Animal Welfare and Ethical Review Body, ensuring compliance with ethical standards and animal welfare considerations. Animals exceeding the tumour size endpoint, defined as an average diameter greater than 15 mm, were humanely euthanized. In this study, no tumours surpassed this upper limit. The maximum allowable tumour burden was determined using the formula (width + length)/$2 \leq 15$ mm. The sample size was determined according to our previous experience and a priori power analysis (G*Power). Housing condition of Gurdon animal facility: all mice used in this study were maintained in specific pathogen free unit on 12 hours light and 12 hours dark cycle. The ambient temperature was 21 °C with a maximum variation of ±2 °C. The humidity was 55 ± 10%. Mice were housed for phenotyping with a stocking density of 3–5 mice/cage in individually ventilated caging provided with standard diet all the time as per UK Home Office's regulation. The mice were genotyped by Transnetyx. *Cyp11a1*-mCherry reporter and *Cyp11a1*ᶠˡ/ᶠˡ mice were generated by Sanger Institute as previously described in Mahata et al., *Nat Commun* 2020. *Cyp11a1*ᶠˡ/ᶠˡ;*Vav1*Cre (Cyp11a1ᶜᴷᴼ) were generated with crossing with *Vav1*-iCre mice (Jackson laboratory). In this study we used all female mice aged between 8 to 12 weeks.

### Haematoxylin & Eosin Staining and CYP11A1 Immunohistochemistry (IHC) Assay
Formalin-fixed paraffin-embedded (FFPE) tissue samples were collected from 22 patients who had undergone either mastectomy or lumpectomy procedures. Tissue sections, meticulously prepared at a thickness of 5-µm, were first deparaffinised using an environmentally friendly dewaxing transparent liquid, and subsequently rehydrated through a graded ethanol series. The sections were then subjected to standard Hematoxylin and Eosin (H&E) staining procedures.

For the Immunohistochemistry (IHC) staining, we employed a high temperature and high-pressure method for antigen retrieval, using Citrate Antigen Retrieval Solution at a pH of 6.0. The sections were treated with 3% $H_2O_2$ for 25 minutes to quench endogenous peroxidase activity, followed by a 30-minute block with 3% BSA at room temperature to minimise non-specific binding. The sections were then incubated overnight at 4 °C with the rabbit anti-human CYP11A1 primary antibody [EPR24868-86] (ab272494), obtained from Abcam, USA, at a 1:200 dilution. Following primary antibody incubation, sections were rinsed with phosphate-buffered saline (PBS) and then incubated with HRP labeled goat anti-rabbit secondary antibody (GB23303) sourced from Servicebio, China, also at a 1:200 dilution. To visualise the antibody-antigen complexes, a DAB solution was applied, and the sections were then rinsed with distilled water. Subsequent counterstaining was performed using hematoxylin. Finally, the sections were dehydrated in an ascending ethanol series, cleared in xylene, and mounted using a gelatin-based medium. The meticulous approach in the preparation and staining of these tissue sections aims to ensure accuracy and reliability in the subsequent analysis of CYP11A1 expression.

### Evaluation of IHC Staining Results
The assessment of the Immunohistochemistry (IHC) staining results was meticulously conducted by two pathologists, who were kept unaware of the patients' outcomes to ensure unbiased interpretation. Employing a semiquantitative scoring system, they evaluated the results based on two main criteria: the intensity (I) of the staining, which ranged from 0 (no staining), 1 (weak), 2 (moderate), to 3 (strong), and the percentage (P) of nuclear staining, categorised from 0 (no expression), 1 (up to 10%), 2 (10–20%), up to a maximum of 100. These two components, intensity and percentage, were then multiplied (I x P) to derive the H-score, providing a comprehensive evaluation of the CYP11A1 expression observed in the tissue samples.

### Multicolour Immunofluorescence Staining
Human tissue sections for multicolour immunofluorescence were processed using a method similar to that of immunohistochemistry

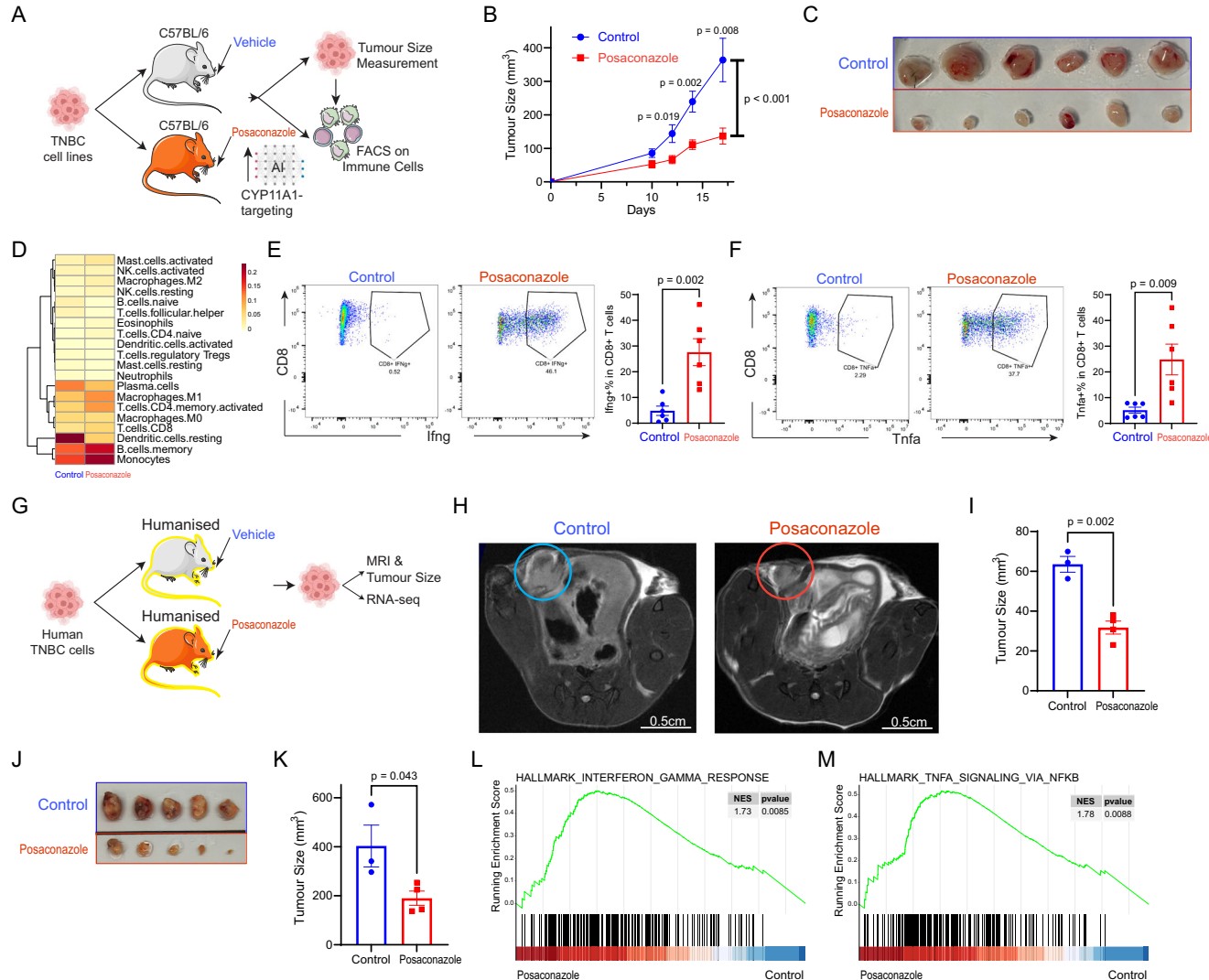

**Fig. 5 | Therapeutic efficacy of posaconazole in TNBC preclinical models.**
**A** Illustration of Posaconazole treatment regimen in a TNBC mouse model.
**B** Longitudinal tumour growth curves in Posaconazole-treated and control mice over time (*N* = 8; (E0771.LMB injected subcutaneously, Mean ± SEM expressed through error bars; a unpaired two-tailed t-test was performed); Source data are provided as a Source Data file. **C** Side-by-side presentation of tumours from both Posaconazole and control groups upon study conclusion (E0771.LMB injected subcutaneously). **D** Heatmap depicting the immune cell populations quantified by Cibersort in tumors from control (left column) and Posaconazole-treated (right column) mice in a TNBC mouse model (E0771 injected orthotopically). **E**–**F** Flow cytometry scatter plots showcasing Ifng (**E**) and Tnfa (**F**) expression in CD8⁺ T cells from Posaconazole-treated and control mice in TNBC mouse model (E0771.LMB injected subcutaneously). Comparative data presented on the right highlights the differential expression across groups (biological replicates, *N* = 6; Error bars denote mean ± SEM; a unpaired two-tailed t-test was performed); Source data are provided

as a Source Data file. **G** Overview of the humanised mouse model setup, indicating the introduction of human TNBC cells orthotopically and subsequent Posaconazole treatment. **H** MRI snapshots after one week of Posaconazole administration, emphasising tumour size differences. **I** Quantitative 3D tumour volume representation derived from MRI scans after one week (Control group *N* = 3, Posaconazole-treated group *N* = 4; Mean ± SEM expressed through error bars; a unpaired two-tailed t-test was performed); Source data are provided as a Source Data file. **J** Photograph of harvested tumours from both groups at the 21-day mark. **K** Quantitative analysis reflecting tumour size disparities post 21 days (Control group *N* = 3, Posaconazole-treated group *N* = 4; Mean ± SEM expressed through error bars; a unpaired two-tailed t-test was performed); Source data are provided as a Source Data file. **L**–**M** GSEA results of indicated gene sets between Posaconazole-treated and control groups (The calculation of GSEA results using clusterProfiler (version 3.18.1) was included in Methods). Figure 5A, H were partly generated using Servier Medical Art, licensed under a CC BY 4.0 license.

(IHC). Antigen retrieval was conducted with Tris-EDTA buffer (pH 8.0). The sections were then incubated overnight at 4 °C with a mouse anti-human CD45 primary antibody (GB115428-100, diluted 1:200, Servicebio) and a rabbit anti-human CYP11A1 primary antibody (EPR24868-86, catalog ab272494, Abcam, USA). Following the primary antibody incubation, sections were treated with the respective secondary antibodies: a CY3-labeled goat anti-rabbit IgG (H + L) (catalog GB21303, Servicebio) and an Alexa Fluor 488-labeled goat anti-mouse IgG (H + L) (catalog GB25301, Servicebio). Nuclei were counterstained with DAPI (catalog G1012, Servicebio) to visualise nuclear morphology and reduce tissue autofluorescence. After staining, the slides were

mounted with an anti-fade mounting medium to preserve fluorescence intensity and prevent photobleaching. The slides were then examined using an Ortho-Fluorescent Microscope (NIKON ECLIPSE C1) equipped for fluorescence detection.

### In Vitro Human DCs Culture and Differentiation

Fresh peripheral blood mononuclear cells (PBMCs) were isolated from the buffy coats of blood donors using a previously described method[75]. Monocytes, enriched to >96% CD14+ purity, were obtained using CD14-microbeads from Miltenyi Biotech (Auburn, CA, USA). Over a period of 7 days, these monocytes were cultured in RPMI 1640 medium

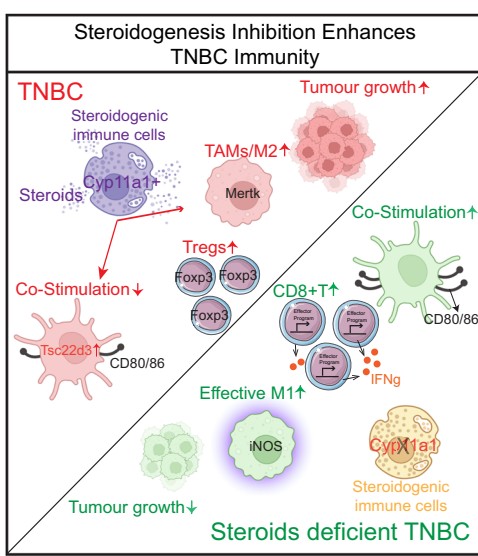

**Fig. 6 | Diagrammatic Illustration of the Discovery.** The graphical illustration shows how local steroidogenesis affects anti-tumour immunity in TNBC. In the TNBC environment (left), steroid-producing immune cells promote tumour growth by fostering immunosuppressive elements like Tregs and tumour-associated macrophages (TAMs). In contrast, steroid inhibition (right) boosts anti-tumour immunity, enhancing CD8⁺ T cell and M1 macrophage activity, and restoring co-stimulation in DCs, which limits tumour growth. This suggests targeting steroidogenesis as a potential therapeutic strategy in TNBC.

supplemented with 10% heat-inactivated FBS, 50 ng/ml GM-CSF (Peprotech, London, UK), and 10 ng/ml IL-4 (Peprotech, London, UK)[76], leading to the dendritic cell (DC) differentiation. Two conditions were established: DC_GC, where 200 nM cortisol (Sigma-Aldrich Merk) was added, and DC_Mock, with no cortisol addition. To induce DCs maturation, 100 ng/ml LPS was administered 24 hours before the cells were harvested.

### Enzyme-Linked Immunosorbent Assay (ELISA)
The levels of human cytokines IL-12p70, IL-6, IL-1β, and IL-10 in the supernatants were quantified following a 7-day culture period of human DCs, which had been subjected to treatments with either glucocorticoids (GC) or a vehicle control. For this quantification, ELISA MAX kits supplied by BioLegend were utilised, and all procedures were conducted in strict accordance with the instructions provided by the manufacturer.

### Allogeneic Mixed Lymphocyte Reaction (MLR) Assay
Human naive T cells were isolated from fresh PBMCs utilising the human naive pan T cell isolation kit provided by Miltenyi Biotech (Auburn, USA). These isolated T cells were then co-cultured with DCs that had been treated either with GC or a vehicle control, using the same quantities for both treatments. The co-culture was set up with 5 × 10^4 allogeneic naive T cells and DCs at a ratio of 1:10, all placed in a complete medium. This setup was maintained for a period of 5 days. Upon completion of the culture duration, intracellular cytokine levels in T cells were meticulously determined using flow cytometry.

### Peritoneal Mast Cells (PMC) isolation and culture
We used 8 to 12-week-old female C57BL/6 N mice for peritoneal mast cell (PMC) isolation. After culling, 7 ml of ice-cold sterile Tyrode's buffer (Thermo Scientific, Cat. No. J67607.K2) and 3 ml of air were injected in the peritoneal cavity using a 10 ml syringe equipped with a 27 G needle. After injection, we shook the mice for 1 min to detach peritoneal cells into the Tyrode's buffer. We aspirated the fluid from

the abdomen gently and slowly (~0.5 ml/s) to avoid clogging by the inner organs and transferred the collected cell suspension into a collection tube. Next, we centrifuged the tubes with the cell suspension at ~300 x g for 7 mins at 4oC. Under a sterile hood, we aspirated the supernatant and washed cells with DMEM complete medium. We resuspended the pellets in 4 ml of pre-warmed DMEM complete medium with IL-3 (20 ng/ml) and Stem Cell Factor (SCF, 30 ng/ml), transferred the cell suspension to a 25 cm2 culture flask and incubated for 12 days (37 °C and 5% CO2). On 2nd and 4th day we changed the medium and transferred the cell suspension on to a new culture flask and continued the culture for 8 more days. On 12th day we analysed their maturation (cKit and FceR1 expression) by flow cytometry.

### Mast Cell Cyp11a1 Inhibition Assay
Matured PMCs were seeded with equal density (200,000 in 200 µl) in 96 well U bottom plate in the presence of Posaconazole (100 uM, 50 uM and 25 uM) or vehicle (DMSO) in pre-warmed IMDM complete medium (without phenol red, 10% charcoal-stripped FBS), and incubated (37 °C and 5% CO2) for 24 hours. After 24 hours the cell supernatants were analysed by quantitative pregnenolone ELISA following manufacturers protocol (Pregnenolone ELISA kit, Abnova). Absorbance was measured at 450 nm, and data was analysed in GraphPad Prism 9. We checked cell viability by flow cytometry.

### Mouse TNBC Tumour Model
The E0771.LMB or E0771 breast cancer cell line was maintained in RPMI 1640 medium (Sigma), enriched with 10% FBS, 1% PenStrep, and 10 mM HEPES. The cells were then injected subcutaneously into wild type or genetically modified mice including wild type (WT), C57BL/6, Vav1^Cre, Cyp11a1^fl/fl;Vav1^Cre, and Cyp11a1-mCherry reporter mice. For the primary tumour growth assay, 5 ×10^5 E0771.LMB or E0771 cells in 100 µl PBS were injected either subcutaneously into the flank region or Orthotopically into the 4th inguinal mammary fat pad. The tumour growth was assessed by taking length and width measurements three times a week starting from day 7 after injection. The tumour volumes were calculated using the following formula (π/6)(shortest length X longest length). After 19 or 21 days animals were killed via cervical dislocation and tissues collected for analysis.

### Tumour Tissue processing
The tumours were mechanically dissociated and were then subjected to enzymatic digestion using a mixture of 1 mg/ml collagenase D (Roche), 1 mg/ml collagenase A (Roche), and 0.4 mg/ml DNase I (Sigma) in IMDM media containing 10% FBS, all incubated at 37 °C for 30–40 minutes. To halt collagenase activity, EDTA (5 mM) was introduced to all samples. The digested tissues were finally strained through 70µm cell strainers (Falcon) to ensure uniformity in sample preparation.

### Posaconazole treatment
Mice received Posaconazole (Noxafil, 40 mg/ml oral suspension) by oral gavage at 20 mg/kg body weight, diluted in water, once daily. In some experiments (subcutaneous), the first dose was administered concurrently with cell injection and continued for 10 days. In others (orthotopic), we administered the first treatment 7 days after the tumours were established. The vehicles were administered orally by gavage in control mice.

### Cell Sorting
Following the processing into single-cell suspensions, the tumour samples underwent a 30-minute incubation at 4 °C with an anti-mouse CD45 (FITC conjugated) antibody, which had been diluted in PBS containing 0.5% bovine serum albumin (BSA). Before sorting we added

DAPI. The stained samples were then sorted utilising either the MoFlo XDP or BD Influx cytometer systems.

## Spectral Flow Cytometry

Cells prepared for flow cytometric analysis were subjected to a 4-hour incubation with PMA (50 ng/ml) and Ionomycin (500 ng/ml), followed by the addition of Monensin (Biolegend) to the culture for the last 3 hours. The established protocols for surface staining and intracellular cytoplasmic protein staining by eBioscience were diligently followed. LIVE/DEAD Fixable Dead Cell Stain (Thermo Fisher) was employed to stain the single-cell suspensions, which were subsequently fixed using the eBioscience IC fixation buffer. For intracellular cytokine staining, the cells were fixed and permeabilised using eBioscience IC Fixation buffer and 1x permeabilisation buffer, respectively. Fluorescent dye-conjugated antibodies were then applied to stain the cells. After a thorough wash with 3% PBS-FCS, the cells were ready for analysis using the Cytek Aurora (5 L) flow cytometer. FlowJo v10.2 software facilitated the data analysis. The panel of antibodies used in combination included Human CD86 (PE), IFNγ (AF-700), and TNFα (PE-Cy7); Mouse CD45 (BV711, BUV563 & FITC), CD11b (APC-Cy7), Mertk (PE-Cy7), iNOS (APC), TCRb (eFluor 450 & FITC), CD4 (APC & PE-Cy7), Foxp3 (PE), CD8 (APC-eFluor 780), IFNγ (PerCp-Cy5.5), TNFα (PE-eFluor 610), FceR1 (FITC), cKit (BV711), SiglecF (PerCP-eFluor 710), along with Live/Dead Fixable Violet (Thermo).

## Orthotopic huPBMC-NOG-dKO Model for Humanised Mouse TNBC

Female NOG-dKO (murine MHC class I- and class II-deficient) mice aged 6 weeks were purchased from Charles River Laboratories. To generate the huPBMC-NOG-dKO mice, peripheral blood mononuclear cells (cat. no. PB009C) were provided by HemaCare Company. The MDA-MB-231 human breast cancer cell line, procured from the public experimental platform at Shandong Cancer Hospital and Institute (Shandong, China), was utilised. Cultivation of the cells was done in DMEM medium (Gibco, Thermo Fisher Scientific), supplemented with 10% fetal bovine serum (FBS, ExCell), 100 units/ml penicillin, and 100 μg/ml streptomycin, with maintenance carried out at 37 °C in a 5% CO2 incubator. Orthotopic injection of $5 \times 10^{\wedge}6$ cells into the 4th inguinal mammary fat pad of the huPBMC-NOG-dKO mice was performed for tumour initiation. Tumour growth was meticulously tracked using daily non-invasive measurements and weekly magnetic resonance imaging (MRI) scans, facilitating the calculation of tumour volumes. Following the euthanasia procedure, tissue samples were swiftly harvested and reserved for RNA sequencing analysis.

## Magnetic Resonance Imaging (MRI) Analysis

MRI procedures were conducted using a 9.4 T Biospec 94/30 Small Animal MRI System (Germany). During the process, the mice were sedated using 1% sodium pentobarbital, ensuring stable respiration before being positioned prone, with their heads placed forward. The imaging process utilised a suitable volume coil for RF pulse transmission and a double-segment surface coil to receive signals. Initial imaging involved the acquisition of orthogonal T1-weighted images for spatial localisation, followed by T2-weighted images captured in both coronal and sagittal planes using a Turbo Spin Echo (TSE) pulse sequence. The procedure's specific parameters were set as follows: Repetition Time (TR) at 2500 ms, Echo Time (TE) at 33 ms, Refocusing Angle at 180°, Number of Excitations at 4, Field of View (FOV) at 25×28 mm², Image Size at 256×256, encompassing 24 slices with each slice having a thickness of 0.9 mm. The entire scanning process for each animal was concluded within a duration of 5 minutes and 20 seconds. Images obtained in DICOM format were further incorporated into the 3D Slicer© (https://github.com/Slicer/Slicer/tree/main) to build the models of three-dimensional (3D) of TNBC tumours.

## Steroid Quantification via Liquid Chromatography Mass Spectrometry (LC-MS/MS)

For tumour tissues from TNBC patients, steroid quantification was carried out via a Liquid Chromatography Mass Spectrometry (LC-MS/MS) system, integrating both UPLC and ESI-MS/MS techniques. Employing high-purity chemicals and reagents, including HPLC-grade acetonitrile, isopropanol, methanol from Merck (Darmstadt, Germany), and MilliQ water, ensured the precision of the experimental setup. Acetic acid from Sigma-Aldrich and various standards from Olchemlm Ltd. were utilised, with the standards prepared at 1 mg/mL in methanol, stored at −20 °C, and later diluted to the required concentrations for analysis.

The sample preparation began with thawing and homogenising the sample, followed by the addition of 400 μL methanol to 0.05 g of the sample. This mixture underwent a series of vortexing, icing, and centrifugation steps, resulting in a supernatant which was then concentrated, redissolved in methanol, and prepared for LC-MS analysis. UPLC analysis was conducted using the ExionLC AD system, utilising a Phenomenex Kinetex C18 column and a specific solvent system, with detailed attention paid to the gradient, flow rate, temperature, and injection volume settings (UPLC, ExionLC AD' https://sciex.com.cn/; MS, QTRAP® 6500+ System, https://sciex.com/).

Mass spectrometry was performed on an AB 6500 + QTRAP® LC-MS/MS System, equipped with an ESI Turbo Ion-Spray interface, capable of operating in both positive and negative ion modes. The instrument settings, including the ion source, source temperature, ion spray voltage, and curtain gas, were finely tuned for optimal performance. Furthermore, multiple reaction monitoring transitions were rigorously set and monitored for each steroids of interest, ensuring accurate and reliable steroid quantification in the samples.

For TNBC tumours and serum samples, E0771 cells, and mast cell culture supernatants: Samples (100 μL) of each cell supernatant sample, was enriched with isotopically labelled internal standards, including 13C2,d2-pregnenolone (1 ng) and extracted along with a mixed steroid calibration curve, including pregnenolone (0.005-1 ng) through supported liquid extraction plates on an Extrahera liquid handling robot (Biotage, Uppsala, Sweden) using dichloromethane/isopropanol (98:2 v/v), reduced to dryness under nitrogen and resuspension in water/methanol (80 μL; 70:30 v/v water/methanol) followed by LC-MS/MS analysis of the extract.

Frozen tissue (~50 mg - exact weight recorded) was placed in 2 mL reinforced tubes (containing 1.4 mm ceramic beads, FisherScientific). Acetonitrile with 0.1 % formic acid (v/v; 1 mL) was added to the tube and it was enriched with 20 uL isotopically labelled steroid standard mixture (as above). Each tube was added to a Bead Ruptor 24 Elite (Omni International) fitted with a CryoCool unit. The tubes were homogenised for 1 m/s for 30 seconds for 3 cycles. The supernatant for each sample was transferred to a Filter⁺ plate (Biotage, Sweden), positive pressure applied and the eluate collected in a clean 96-well collection plate. The filtered homogenate was further processed through a phospholipid depletion (PLD + ) plate (Biotage, Sweden) and the eluate was reduced to dryness, resuspended in water/methanol (70:30 v/v) and the plate sealed with a zone-free plate seal, ready for LC-MS/MS analysis.

Briefly, an I-Class UPLC (Waters, UK) was used for the liquid chromatography on a Kinetex C18 column (150 ×2.1 mm; 2.6 μm) with a flow rate of 0.3 mL/min and a mobile phase system of water with 0.05 mM ammonium fluoride and methanol with 0.05 mM ammonium fluoride, starting at 50% B, rising to 95% B and returning to 50% B. Separation of 18 steroids was carried out. The column and autosampler temperatures were maintained at 50 and 10 °C, respectively. The injection volume was 20 μL and the total analytical run time per sample

was 16 min. Steroids were detected on a QTrap 6500+ mass spectrometer (AB Sciex, Warrington, UK) equipped with an electrospray ionisation turbo V ion spray source. Positive ion spray voltage was set to 5500 V and negative ion spray voltage was set to -4500 V, with the source temperature maintained at 600oC. Multiple reaction monitoring parameters were carried out for all steroids including pregnenolone (P5) m/z 317.1 281.1 and 159.0 with declustering potential (DP) of 66 collision exit potential (CXP) of 31 and 29 V and collision energy (CE) of 12 V, respectively and for 13C2,d2-pregnenolone of 321.2 * 285.2 with DP of 14 CXP of 17 and CE of 18 with retention time of 10.4 mins.

The ratio of P5/13C2,d2-P5 peak areas were calculated and linear regression analysis used to calculate the amount of P5 in each sample. The same was done for other steroids in the sample (aldosterone, progesterone, 17β-estradiol, 5α-dihydrotestosterone and testosterone) by evaluation of the data on MultiQuant 3.0.3 (AB Sciex, UK).

## scRNA-seq Library Preparation and Sequencing

Single-cell RNA sequencing libraries were crafted at the Cancer Research UK Cambridge Institute Genomics Core Facility, employing the Chromium Next GEM Single Cell 5′ Kit v2, along with other specialised kits and guides provided by 10X Genomics. The process began with sorting samples into PBS-0.04% BSA solution, which were then loaded into Chromium microfluidic chips. The chips utilised 10X Genomics' 5′ v2 chemistry to create gel-bead emulsions in a Chromium controller, targeting around 10,000 emulsions with unique cell barcodes.

The RNA from these barcoded cells was reverse transcribed using a C1000 Touch Thermal cycler (Bio-Rad). The cDNA obtained underwent quality and quantity checks using an Agilent TapeStation 4200. Approximately 1000 ng of the cDNA material proceeded to library preparation, with sample indexing cycles tailored due to variance in cDNA amounts. Library quality was ascertained using the Agilent TapeStation 4200 and BMG LABTECH Clariostar Monochromator Microplate Reader, ensuring accurate dsDNA quantification. Post-normalisation to a 10 nM concentration, samples were pooled for sequencing, occupying 16% of a NovaSeq6000 sequencing lane. The sequencing parameters were set for a mix of short and long reads, aiming for a substantial depth per cell to achieve a total of around 2 billion reads. The sequencing output was compiled into FASTQ files, accompanied by a multiQC report detailing the single-cell data.

## Library Construction, Quality Control and mRNA Sequencing

RNA samples from DCs were isolated using RNeasy Plus Mini Kit (Qiagen). Messenger RNA was purified from total RNA using poly-T oligo-attached magnetic beads. After fragmentation, the first strand cDNA was synthesised using random hexamer primers followed by the second strand cDNA synthesis. The library was ready after end repair, A-tailing, adapter ligation, size selection, amplification, and purification. The library was checked with Qubit and real-time PCR for quantification and bioanalyzer for size distribution detection. Quantified libraries were pooled and sequenced on Illumina platforms, according to effective library concentration and data amount (Novogene Co., Ltd).

For tumour samples from humanised mice, total RNA was isolated using TRIzol reagent (Invitrogen, Carlsbad, CA, USA) and assessed for quantity and purity via NanoDrop ND-1000 (NanoDrop, Wilmington, DE, USA). Ensuring RNA integrity, RIN numbers were maintained above 7.0, verified through Bioanalyzer 2100 (Agilent, CA, USA) and agarose gel electrophoresis. Subsequently, poly (A) RNA was purified from 1 µg of total RNA utilising two rounds of purification with Dynabeads Oligo (dT)25-61005 (Thermo Fisher, CA, USA). The RNA was then fragmented and converted to cDNA using Magnesium RNA Fragmentation Module (NEB, USA) and SuperScript™ II Reverse Transcriptase (Invitrogen, USA). This was followed by U-labeled second-stranded DNA synthesis, incorporating E. coli DNA polymerase I, RNase H, and dUTP Solution (NEB and Thermo Fisher, USA). Post A-tailing, adapters were ligated to the fragmented DNA, followed by size selection using

AMPureXP beads and UDG enzyme treatment (NEB, USA). The ligated products underwent PCR amplification, resulting in a final cDNA library with an average insert size of 300 ± 50 bp. Finally, 2×150 bp paired-end sequencing was conducted on an Illumina Novaseq™ 6000 (LC-Bio Technology CO., Ltd., Hangzhou, China), adhering to the manufacturer's recommended protocol.

## The SteroidOmics Integrative Analysis

Steroid hormone concentration data, integrated with publicly available RNA-seq dataset[52], was analysed to scrutinise steroid signals in TNBC. For each steroid under investigation, average steroid concentrations were calculated across patient cohorts, and average gene expression levels were assessed for the associated producing and receptor genes of each steroid, using RNA sequencing data. A pre-defined steroid-gene dataset, derived from the KEGG database (http://www.kegg.jp), was employed to facilitate the identification of pertinent producing and receptor genes for each steroid. Subsequent to the calculation of these average values, a z-score normalisation procedure was applied separately across the patient samples for steroid concentrations, as well as for producing and receptor gene expressions. An overall steroid signal was then defined through the computation of the row mean of the z-scores corresponding to the steroid concentrations and gene expressions.

## ScRNA-seq Data Processing

The raw single-cell RNA sequencing (scRNA-seq) data were subjected to an initial preprocessing stage employing the Cell Ranger toolkit (version 7.1.0), a product of 10x Genomics. This stage involved the filtration of low-quality reads, alignment of reads to the mouse reference genome (GRCm38), demultiplexing of cellular barcodes, and generation of the unique molecular identifier (UMI) matrix. Gene expressions observed in fewer than three cells were subsequently eliminated from the expression matrix. In terms of basic quality filtering, cells exhibiting a gene count greater than 200 but fewer than 7,500, alongside a UMI count below 60,000, were retained. Additionally, cells displaying more than 20% of their genes as mitochondrial were discarded. The resultant filtered matrix underwent normalisation using the LogNormalize method in Seurat (version 3.2.2)[77], setting the scale.factor to 10,000. Concurrently, Percent.mt and nCount_RNA were regressed out during the scaling step.

The scRNA-seq data derived from both tumour tissues and matched blood samples of TNBC patients utilised across all studies are publicly accessible via the Gene Expression Omnibus, under the accession number GSE169246 and PRJEB35405[53]. For the dataset originating from Zhang et al., cells displaying fewer than 400 or more than 8,000 genes, a UMI count below 600 or exceeding 120,000, or a mitochondrial gene percentage greater than 10% were excluded. Post these filtration steps, all samples underwent integration using Seurat (version 3.2.2) to mitigate batch effects, followed by normalisation, employing the aforementioned method.

## Unsupervised Clustering and Annotation of Cell Types

To discern highly variable genes conducive to unsupervised cell clustering, Seurat (version 3.2.2) was employed. Subsequently, a principal component analysis (PCA) was executed on the leading 2,000 highly variable genes. The determination of the significant principal components (PCs) for downstream analysis was informed by the ElbowPlot function of Seurat, settling on 20 significant PCs. Employing the FindNeighbors function in Seurat, a Shared Nearest Neighbor (SNN) Graph was constructed. The subsequent unsupervised clustering was executed using the Seurat function FindClusters, with a resolution parameter set to 0.4. These principal components were further harnessed for non-linear dimension reduction, thereby creating visual projections through the Uniform Manifold Approximation and Projection (UMAP) or t-Distributed Stochastic Neighbor Embedding (t-SNE) using Seurat functions RunUMAP and RunTSNE respectively.

For the purpose of annotating cell types and states, the Seurat toolkit (version 3.2.2) was utilised to pinpoint differentially expressed genes (DEGs) across various cell clusters, applying the FindAllMarkers function. Subsequent to this analysis, both the top 50 most significantly regulated DEGs and gene signatures characteristic of principal immune populations were meticulously examined. Regarding the dataset from Zhang et al., previously reported methods were adopted for cell annotation[53].

## Gene Set Score Calculation

In the process of quantifying the activity of specific signalling pathways, we utilised curated gene sets labelled as "Gobp steroid biosynthetic process", "Gobp response to steroid hormone", "Reactome costimulation by the cd28 family", "Gobp antigen processing and presentation", "Hallmark inflammatory response", and "Gobp mast cell activation", all sourced from the Molecular Signatures Database (MsigDB) (http://www.gsea-msigdb.org/gsea/msigdb/index.jsp). Additionally, we incorporated signature gene lists for M1 and M2 macrophages derived from the LM22 matrix[78], a T cell exhaustion gene list as defined by Zhang et al.[53], as well as genes associated with dendritic cell maturation and tolerance according to Robertson et al.[79]. Following the compilation of these gene sets, scores for each were computed employing the AddModuleScore function in Seurat (version 3.2.2), utilising the function's default parameter settings.

## Single-cell Flux Balance Analysis

For the derivation of single-cell metabolic flux matrix pertaining to immune cells from TNBC tumours, the Compass method, a validated approach for single-cell Flux Balance Analysis (scFBA), was employed[56]. Core metabolic reactions within both our scRNA-seq datasets and TNBC patient datasets were characterised, utilising the reaction metadata provided by the Recon2 database. Comprehensive documentation and instructions for implementing Compass can be accessed via GitHub (https://github.com/YosefLab/Compass). Subsequent to the acquisition of the single-cell flux matrix, it was integrated into a Seurat object, employing the Seurat R package, to facilitate subsequent visualisation and analysis.

## Bulk RNA-seq Analysis

Quality assessment of the raw sequencing reads was conducted using FastQC, and the HISAT2 aligner was utilised to map the reads to the human genome (hg38), resulting in SAM files for each sample. Samtools was subsequently used to convert and sort these files into BAM format. Gene-level abundance and raw counts were derived using htseq-count, providing the necessary data for principal component analysis (PCA) and differential gene expression analysis in DESeq2 (version 1.30.1)[80]. Gene set enrichment analysis (GSEA)[81] were performed with clusterProfiler (version 3.18.1)[82].

To assign immune cell phenotypes within the TME for deconvolution analysis, the CIBERSORTx web portal (https://cibersortx.stanford.edu/runcibersortx.php) was utilised. The gene expression matrix derived from TNBC patients was inputted into the CIBERSORT tool. For this analysis, the LM22 signature matrix, which includes gene expression profiles for established immune markers, was employed as the default cell-type signature matrix. Correlations and p-values were calculated to explore the relationships between each steroid hormone receptor genes and the different immune population scores derived from CIBERSORT. This involved a systematic iteration over the selected genes, during which the Pearson correlation coefficient was computed for each gene in relation to the immune cell score. To assess the statistical significance of these correlations, p values were determined. Following this, adjustments were made to the p values to ensure accurate interpretation of significance levels and bolster the reliability of the results.

## Gene Expression Visualisation

Specific gene set scores and gene expression were visualised using heatmaps, violin plots, and bar plots, generated with the pheatmap package (version 1.0.12), available at (https://cran.r-project.org/web/packages/pheatmap), and the dittoSeq R package (version 1.2.4)[83]. During the visualisation process, data was automatically scaled using the default parameters provided by these packages.

## Statistics and reproducibility

For the calculation of DEGs in single-cell/bulk RNA-seq datasets, p-values were derived using the default methodologies embedded in the respective R packages. The ggpubr R package (version 0.4.0), accessible at (https://CRAN.R-project.org/package=ggpubr), was employed to compare differences in specific gene set scores and metabolic flux levels across immune cells from varying groups.

In both in vitro and in vivo experimental settings, Prism9 (GraphPad Software) was utilised to ascertain significance levels of the data, applying a two-tailed Student's t-test and a one/two-way ANOVA for analytical purposes. A p-value falling below the 0.05 threshold was deemed to indicate statistical significance.

In all experiments, N defines the cohort size or independent biological replicates unless stated. Each representative experimental result displayed represents at least two (Figs. 2G, K, L, 3M, N, 5K) or more (Figs. 1F, G, H, 2C, 5D) independent experiments. scRNA-Seq and steroid profiling (LC-MS/MS) experiments were done only once with 3-16 Independent biological replicates. Summary results were shown in Figs. 2B, 3F, G.

## Reporting summary

Further information on research design is available in the Nature Portfolio Reporting Summary linked to this article.

## Data availability

Newly generated RNA sequencing data of this study is submitted to NCBI Gene Expression Omnibus (GEO) with the accession number at GSE255495. The scRNA-seq and RNA-seq data derived from tumour tissues and blood samples of TNBC patients reused in this study are publicly accessible via the GEO and Sequence Read Archive, under the accession number GSE169246, PRJEB35405, and SRP157974. Source data are provided with this paper.

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

## Acknowledgements

We would like to thank Joana Cerveira and Richard Grenfell for help with flow cytometry; Louise van der Weyden for providing E0771.LMB cell line; UBS animal facility, Gurdon Institute, for their technical help and animal husbandry; Zhiqiang Shi for the collection of clinical samples. Some figures (Figs. 1A, 2A, 3D, 4A, 4H, 5A, 5H, and S1C) were partly generated using Servier Medical Art, licensed under a CC BY 4.0 license. The work is supported by CRUK Career Development Fellowship (RCCFEL\100095), NSF-BIO/UKRI-BBSRC project grant (BB/V006126/1), MRC project grant (MR/V028995/1), CRUK Cambridge Centre Cancer Immunology Programme Pump Priming award, and CRUK CC MRes/PhD Studentship.

## Author contributions

Q.Z.: Act as a project lead. Conceptualised, designed and performed experiments. Analysed, assembled, and visualised data. Wrote the manuscript. J.P.: Designed and performed experiments, analysed data, and helped Q.Z. Y.L: Collected clinical samples and performed experiments. NH: Involved with optimisation of samples, steroid detection, and mass-spectrometry data analysis. C.I. and B.Z.: Contributed to the mice experiments that involved subcutaneous injection and oral gavage. R.R. and K.O. supervised in vivo aspects of the study. M.I.: Prepared PBMCs for the study. A.C.M.: Obtained and maintained ethical approval for healthy donor PBMC isolation. Supervised MI and reviewed the manuscript. S.K.S.: Helped in tissue processing. P.Q.: Conducted and supervised humanised mice experiment, patients' steroid analysis and CYP11 A1 IHC. B.M.: Led and managed the team. Supervised the study. Reviewed the manuscript. Involved with conceptualisation, experimental designing, fund acquisition and resource management. All authors commented on and approved of the draft manuscript before submission.

## Competing interests

BM, JP and SKS declare following competing interests. An UK patent application submitted (Title: Cancer Treatments. Reference Number: P370182GB. Patent Application Number: 2502017.3) partly based on the findings in this manuscript where BM, JP and SKS are co-inventors. All other authors declare no competing interests.
