## [Transparent Peer Review file · Nature Communications]

Perturbing local steroidogenesis to improve breast cancer immunity

Corresponding Author: Dr Bidesh Mahata

Version 0:

Reviewer comments:

Reviewer #1

(Remarks to the Author)

This is very interesting paper on the important subject how to improve the immune responses against tumor by modulating local steroidogenic activities.

The subject is important, and its content would benefit from better coverage and recognition of other work on extra-adrenal and extragonadal steroidogenesis in peripheral organs, see citation (Genes Immun 21(3):150-168, 2020. doi: 10.1038/s41435-020-0096-6).

Otherwise, the paper for the most part is well written. The methodology is sound, and data properly collected and interpreted. I do not have any major critique in this area.

However, the microscopic bars in images are almost invisible. This requires corrections.

Also, arrows pointing to stains of interest would be appreciated by the readers.

As relates to the Cyp11A1, in the discussion it is suggested to acknowledge that in addition to serving as a rate limiting enzyme of steroidogenesis it can activate, vitamin D3, lumisterol and tachysterol to biologically active metabolites as briefly reviewed in (Journal of Investigative Dermatology. 2023, 143(12): 2340-2342. doi: <https://doi.org/10.1016/j.jid.2023.07.003>).

In this context local expression of this enzyme could play a positive homeostatic role.

Again, the subject of local steroidogenesis within tumor deserves more consideration in a wider manner because of its clinical implications and implications for natural history of cancer.

In this context mentioning that tumor can affect its environment and body homeostasis when advanced as discussed recently (How cancer hijacks the body's homeostasis through the neuroendocrine system, Trends Neurosci 46: 263-275, 2023) would be appreciated by the readers.

In summary, this is an interesting study that should be of strong interest to the readers of Nature Communications, after minor revisions as suggested above.

Reviewer #2

(Remarks to the Author)

The major aim of this study, entitled "Perturbing local steroidogenesis to improve breast cancer immunity", was to evaluate the role of steroids in TNBC progression, exploring the possibility that steroids accelerate tumor growth by remodeling the immunosuppressive activity of the tumor microenvironment (TME). To this end, the authors combined multi-omics analysis (metabolomics, transcriptomic) to characterize steroidogenesis in TNBC tumors, and subsequently, conduct genetic and pharmacological perturbation studies, using conditional knockout of Cyp11a1, the rate-limiting enzyme of steroidogenesis, and the newly identified steroidogenesis inhibitor Posaconazole. The authors then convincingly showed in animal models that these perturbations significantly reduced tumor growth and decreased its volume. Using several functional assays and transcriptomic profiling the authors showed that steroidogenesis blockades affect immunosuppressive of tumor associated macrophages (TAMs) and enhanced anti-tumor immune response of dendritic and T cells.

Based on further analysis (scRNAseq, tumour steroidomics) the authors suggest that mast cells and basophils are the predominant steroidogenic agents in TNBC for enhanced therapeutic precision.

Overall, it is a comprehensive study involved in multiple omics data and animal models, describing significant and promising

effects on tumor growth. While these parts are convincing, the others are confusing and primarily descriptive, lacking a clear mechanistic data of how inhibition of steroidogenesis indeed attenuates tumor growth. There are many correlations and supporting evidence but not direct experimental data of mechanisms. Dealing with macrophage, T cells, dendritic cells, mast cells and basophils; it is unclear how all these works together and how to dissect the contribution of each for the end point outcome on tumor growth.

Specific points

(1) Figure 1: Functional Steroidogenesis.

(1A) The authors performed proteomics of 16 tumors from TNBC patients to characterize their steroidogenesis. The figure should include information of their molecular subtyping, at least if they are LAR (luminal androgen receptor) positive.

(1B) It is unclear whether the profiling described in the figure is characteristic to TNBC as it was performed for only 16 samples. There are many studies describing metabolic data of TNBC patients, using much larger cohorts. For example- 10.1038/s41422-022-00614-0. Perhaps the authors can gain supportive evidence from larger cohorts of previous studies.

(1C) The data is shown on TNBC. Is it specific for TNBC? what about non-TNBC?

(1F) Combining IF analysis to the IH data to show by co-staining the identity of CYP11A1-positive cells, at least with immune marker (CD45 positive).

(S1D) It will be more convincing to show FACS analysis of M2 macrophage markers and NR3C1.

(2) Figure 2: Deletion of Cyp11a1 in Immune Cells Restricts TNBC Tumour Growth

(2A) Cyp11a1 KO should be shown by genetic analysis and by WB

(2B) EO771.LMB implanted subcutaneously- orthotopic model would possibly be more suitable.

EO771.LMB is metastatic clone of EO771, very different from EO771, which is usually used as a model for primary tumor- why this cell line was selected?

(2F) The results showed significant upregulation of NK cells- any impact?

(2I) MertK was used as a marker for M2; other canonical markers CD206/163/FIZZ1 are usually used.

(3) Figure 3: Steroidogenesis Inhibition Augments Anti-Tumour Immunity

(D-I) in vitro analysis of monocytes isolated from peripheral blood \pm GC treatment highlights possible role of immunoregulatory cytokines such as IL-10, TGF β 1 etc that might have impact on the tumor growth. Will be important to look on monocyte (DC) from the tumors (control and KO) and evaluate the expression levels of these cytokines. Will be also important to define the ratio between M1 and M2 macrophages from in vivo tumors.

(4) Figure 4: Characterization of Cyp11a1+ Immunocytes in TNBC

The author suggest that mast cells are the predominant population that "modulates immune response in TNBC by contributing to the local immunosuppressive steroid biosynthesis (Figure 4D, 4F). The immune suppressive capacity of mast cells is attributed by the secretion of IL-10, TGF β , prostaglandin D2's that skew dendritic cell function driving a Th2 immune response".

While this is a feasible possibility, the data presented in figure 4 doesn't prove it, only provides some correlative findings, including data shown in Fig. 4G, which could be not related to the presented study. Demonstrating the role of mast cells as key mediators of steroidogenesis associated TNBC progression is required.

(5) Figure 5: Posaconazole Diminishes TNBC Progression

Characterizing the immune landscape of control and Posaconazole-treated tumors is required, considering all the above findings, including M1/M2 macrophage ratio, mast cells etc.

(6) A model/scheme describing the major findings could help

Reviewer #3

(Remarks to the Author)

Immunotherapy in breast cancer has limited efficacy, with the most promising response being observed in triple negative breast cancer (TNBC), a highly aggressive form of the disease. Identifying mechanisms of immune suppression in TNBC should provide avenues for improving response to immunotherapies. Given the ability of glucocorticoids to suppress immune function, this study aims to identify whether local steroid hormone production may impact infiltrating immune cells in TNBC. Using LC-MS/MS analysis of human TNBC samples, they identify a set of steroids that are present in tumors. Using a larger cohort of patient samples, they further demonstrate that immune cells express the mRNAs for a variety of steroid receptors, suggesting that immune cells could respond to steroids in the local microenvironment. Immunostaining for the rate limiting step of steroid biosynthesis, CYP11A1/p450scc, revealed high expression in the immune component of tumors and that immune cells in the tumor have a high steroid biosynthesis score. Thus, the authors ask whether steroid production by immune cells can impact tumor immunity and if this activity can serve as a therapeutic target. Supporting this concept, targeted disruption of Cyp11a1, which will reduce steroid production, in immune cells suppresses TNBC growth. It also shifts the immune composition of tumors. Direct treatment of dendritic cells in vitro with cortisol shifted their cytokine profile. Single cell RNA-seq data further suggested that basophils and mast cells have increased likelihood for producing pregnenolone, the derivative of cholesterol produced by CYP11A1. Evaluating patient data demonstrated that increased mast cell number was associate with worse patient outcomes. Lastly, to demonstrate the importance of CYP11A1 in tumors, a humanized mouse model of TNBC was treated with posaconazole, an inhibitor of CYP11A1, and the growth of tumors was suppressed and this was associated with changes in the expression of cytokine signature gene sets.

While immune cell production of steroids is an intriguing regulator of tumor immunity. There are several major weaknesses with the current study that limit its impact.

1) There is limited novelty in the study.

A) The senior author published a Nature Comm. study in 2020 indicating that steroid biosynthesis in T-cells was important for tumors to suppress immunity using many of the same approaches, but in a melanoma model.

B) The authors also report that posaconazole is a “newly identified drug” that inhibits steroidogenesis. This drug was FDA approved in 2006 and reported to be an inhibitor of CYP11A1 in 2013 (Mast, et al. Mol. Cell. Endocrinol.). It is highly related to ketoconazole, which has been used for decades to block steroid production. It has been shown to inhibit the growth of many tumor types.

C) It is well established that glucocorticoids will impact dendritic cells in vitro. The analysis shown in the current manuscript is not particularly informative.

D) In contrast to the TIMER RNA-based analysis shown in this study, mast cells have been examined in a tissue microarray of a large cohort breast cancer patients (4,444 cases) and associated with good outcomes. This is a major discrepancy and that the authors should address.

2) There is no consideration of whether immune cell production of steroids has any impact on local concentrations of steroids. If not, the authors can only conclude that CYP11A1 function is important for immune cell function, but not that steroids are involved. Do tumor cells also produce steroids? More importantly, are the steroids that are produced by the ovary and adrenal gland, and certainly in the local environment of the tumor, higher than what is produced by the immune cells? These issues should be addressed because it is not clear whether inhibition/knock-out of CYP11A1 cause changes other than steroid levels, such as accumulation of cholesterol that could shunt this key metabolite to other pathways, ultimately impacting immune cells. Direct assessment of local steroid levels in the immune KO of Cyp11a1 is needed to demonstrate a steroid-based immunomodulatory effect.

3) The systemic drug treatments cannot be used to make conclusions regarding immune cell production of steroids because these treatments will change circulating levels of the same steroids.

4) The authors focus on glucocorticoids to the exclusion of androgens and progestins based on an RNA analysis. They should directly assess the impact of these other steroids to warrant conclusions regarding glucocorticoids. In addition, there is a failure to discuss the highly controversial roles of glucocorticoids in breast cancer that have been previously reported. In some cases, glucocorticoids have been reported to be tumor promoting and in other cases, tumor suppressing in breast cancer.

Minor comments:

1) The mast cell survival analysis data are unrelated to local production of steroids. Thus, their contribution to the study is marginal.

2) The treatment protocol for posaconazole is not included. What was the dose, the dosing schedule, and the route? Most importantly, what was the size of tumors prior to treatment or was treatment initiated when tumor cells were implanted? If the drug was given at the time of tumor cell implantation, this has no relevance to human disease.

3) There is inconsistency about the site of the xenografts. In some cases “subcutaneous” is stated and in others “orthotopic”. Were the tumor cells injected directly into the mammary fat pad?

4) Posaconazole has also been reported to inhibit CYP46A1.

Reviewer #4

(Remarks to the Author)

Introduction

- Would benefit from more explanation of what steroid hormones are, and whether you are referring to corticosteroids or sex steroids (like in the Mahata Nature Comms paper <https://pubmed.ncbi.nlm.nih.gov/32680985/>)
- Line 61-62 – would not say ‘limited treatment avenues’; there are many treatments available, however they are not targeted options and have suboptimal clinical efficacy. Recommend more recent citations here – a lot has changed since 2010.
- Line 73 – “principal drivers” might be a little strong – macrophages are important but not the ‘principle’ driver of tumour growth and metastasis
- Line 92 – I think “responsive to steroid hormones” needs to be explained further – does responsive mean encourages or discourages growth? Also I think it would be useful to say corticosteroids here, TNBC is by definition not responsive to sex steroids
- Line 114 - What does ‘analytic experimental assays’ mean?
- Line 124 – posaconazole is not newly identified – it’s an antifungal that’s been used clinically for almost 20 years. Do you

mean newly identified in use for inhibiting steroidogenesis?

Results

Functional Steroidogenesis and Steroid-Signalling Exist in TNBC Tumours

- scRNA-seq has shown steroidogenesis signature in immune cells compared to peripheral blood, but it is possible to tell if the tumour cells themselves are also synthesising steroid?

Genetic Deletion of Cyp11a1 in Immune Cells Restricts TNBC Tumour Growth and Alters Immune Infiltration in the TME

- This Cyp11a1 null mouse model is a nice experiment. You show that there are differences in the immune infiltrate in tumours, but is there a difference in the systemic immune profile of cells? Is it possible that the Cyp11a1 is influencing the production or maturation of immune cells and this is why you see a shift? On the other hand, I find this experiment extremely hard to interpret – how can it be that profoundly altering steroidogenesis across all immune types (which could be the same as completely disrupting immune function) inhibits tumor growth. Why do we not see an acceleration of tumor growth when ALL immune cells are impacted? What happens if you put wild type macrophages back into the CYP11Ako mice. Do the tumors grow again? Did you measure neutrophils? I can't see them in figure 1E or F? I find this odd. Figure 2I-J – are there fewer total immune cells in these mice? This has not been properly quantitated.

- Critically, I don't understand this experiment in the context of the authors own previous work (Nat Comm 2020, showing that specifically inhibiting CYP11A in Tcells is enough to inhibit TNBC tumor growth? In that paper, the authors show that inhibiting CYP11A in Tcells is enough to curb TNBC tumor growth? This work contradicts the current findings of this paper suggesting that it is macrophage specific Cyp11A1 that is critical in TNBC. At the very least, those findings need to be properly discussed at the beginning of this paper.

- The application of exogenous glucocorticoid to DCs and the resulting immunosuppressive phenotype (Fig 3E-J) is not new.

- Having trouble understanding why preventing endogenous steroid production protects against the effects of exogenous steroid – do you have an explanation of the mechanism here?

- For all facs plots – it looks there are always much fewer cells in the treated/Ko situations – have the authors corrected for this? At the moment it could be that the reduction presented are due to overall fewer numbers of cells (and it is not clear in the text – figs 2I, J, fig 3 H, I)

- Critically, the authors have not addressed why the most significant change in the CYP11Ako mouse is a reduction in macrophages, yet all of figure 3 is focused on GC treatment of dendritic cells. Why the switch in cell type? This is really unclear. What is the effect of CYP11A1 ko, and GC treatment on macrophages?

Characterization of Cyp11a1+ Immunocytes in TNBC for Enhanced Therapeutic Precision

- The focus on mast cells is again odd. In previous work (see above) the authors suggest it is Tcells. In this paper it is now potentially mast cells, DCs or macrophages.

- Mast cells identified as responsible for steroidogenesis (insufficient evidence in human samples for basophils)

- Line 273 – careful with wording, posaconazole is not new

Newly Identified Drug Posaconazole Diminishes TNBC Progression in Preclinical Models

- Your results in Fig 5 are nice but I am not convinced that this is due to posaconazole inhibiting CYP11A1

- Importantly, what are the effects of the posaconazole on Mast cells, DCs, macrophages as per the previous figures?? Why are Tcells now effected? These results are very confusing.

- Broadly – posaconazole (and all azole antifungals) are dirty drugs with innumerable off-target effects. There is already a lot of research looking into repurposing antifungal drugs in cancer across an array of targets including metabolism, angiogenesis, stroma, signalling pathways like AKT and mTOR.

- I'd encourage you to consider the safety of antifungal drugs in cancer patients

(<https://academic.oup.com/jac/article/65/3/410/745723>)

- This drug is known to lead to adrenal insufficiency in humans, requiring administration of dexamethasone, a GC agonist to mitigate!

Discussion

- Line 377 – your statement relating to ref 54 is incorrect. This study is about antifungal prophylaxis not treatment, and while posaconazole reduced the rate of invasive fungal infections it did not improve survival in the overall population

- I feel that the information about steroid biosynthesis within immune cells is interesting and potentially targetable, but I don't think that this is adequately linked with the effects of posaconazole, nor do I think that these experiments justify a clinical trial of posaconazole in TNBC

Version 1:

Reviewer comments:

Reviewer #1

(Remarks to the Author)

The authors adequately revised the manuscript, which should be of strong interest to the readers.

There are some problems with reference formatting. For example, reference 70 miss volume and pages number, while in others the name of the journal is not capitalized. However, these can be easily corrected at the proofs stage.

Reviewer #2

(Remarks to the Author)

The authors improved the manuscript and addressed most of the questions of the reviewers.

I dont have further questions.

Reviewer #3

(Remarks to the Author)

The authors have adequately addressed this reviewer's concerns.

Reviewer #5

(Remarks to the Author)

I have reviewed the revised manuscript, specifically focusing on responses to Reviewer #4's comments. Most concerns were addressed, and the manuscript improved.

I do have additional questions and comments regarding the potential clinical application of drugs inhibiting steroidogenesis, including posaconazole. Does the planned phase II clinical trial involve administering posaconazole in a neoadjuvant setting? Most initial phase II trials are designed in the metastatic setting, where the patients enrolled will have received previous treatment, especially if it is for TNBC. Because the TNBC samples used in this study were from treatment-naive patients, the TME, including the immunosuppressive environment and steroidogenesis, may not reflect the heavily treated or actively progressive tumors in the metastatic setting. I suggest a careful statement in the summary paragraph regarding the "clinical trial phase" or "optimistic" precise and effective treatment, which cannot be concluded only from a planned phase II trial that has not been registered yet.

An important biomarker in TNBC that reflects subtypes within TNBC is the androgen receptor. Despite the study's main findings regarding corticosteroid hormones, the results on Androgens in Figure 1D about the integrated metabolomics data on steroid hormone concentrations with transcriptomic data on genes and receptors related to steroids are intriguing. It would be interesting to segregate the clinically AR-positive and AR-negative samples for analysis and observe the overall score to confirm that corticosteroids have higher scores.

I have some minor comments about some of the editing errors noted in the revised portions of the manuscript.

1. pg 4-5. lines 114-127. Glucocorticoids are used instead of GC, and I think you are referring to glucocorticoid receptor with GR? It has not been abbreviated before this paragraph. Also, there are numerous places throughout the manuscript where glucocorticoid is used instead of GC.
2. pg. 7. lines 209-210. The sentence about Figure S2B is unnecessary, as is the repeated figure in Figures 2D and S2B.
3. pg. 10. lines 308-311. The added explanation about posaconazole alters the wording about the posaconazole Cyp11a1 activity. I suggest rewriting the sentence for clarity.
4. Figure 5D needs labeling or explanation about the two columns.
5. I consider posaconazole a generic name and thus would not use capital letters throughout the manuscript. It is currently used in both uppercase and lowercase letters.

Reviewer 1

Points	Reviewers comment	Our response
1.01	This is very interesting paper on the important subject how to improve the immune responses against tumor by modulating local steroidogenic activities. The subject is important, and its content would benefit from better coverage and recognition of other work on extra-adrenal and extragonadal steroidogenesis in peripheral organs, see citation (Genes Immun 21(3):150-168, 2020. doi: 10.1038/s41435-020-0096-6). Otherwise, the paper for the most part is well written. The methodology is sound, and data properly collected and interpreted.	We sincerely thank the reviewer for their generous appreciation of our work and for taking the time to provide valuable suggestions on our manuscript. We have discussed those important findings to put our new research findings into the context. Also cited those seminal works in the revised manuscript. It appears as follows: “Steroidogenesis (steroid biosynthesis) is a biosynthetic process by which cholesterol is converted to steroid hormones. The biosynthesis of steroids starting from cholesterol is often termed de novo steroidogenesis. Cytoplasmic cholesterol is imported into the mitochondria, where the rate-limiting enzyme CYP11A1 (also known as P450 side chain cleavage enzyme) converts cholesterol to pregnenolone. Pregnenolone is the first bioactive steroid of the steroidogenesis pathway, and the precursor of all other steroids. The steroidogenesis pathway has been extensively studied in the adrenal gland, gonads, and placenta. De novo steroidogenesis by other tissues, known as extra-glandular steroidogenesis, in brain, skin, thymus, and adipose tissues, has also been reported. Steroid production because of immune response in the mucosal tissues, such as in the lung and intestine, has been shown to play a tolerogenic role in maintaining tissue homeostasis.” Thank you for appreciation.
1.02	However, the microscopic bars in images are almost invisible. This requires corrections. Also, arrows pointing to stains of interest would be appreciated by the readers.	We thank the reviewer for this suggestion. We have regenerated the scale bars to ensure they are visible and added arrows to point out stains of interest in Figure 1F.

		F 1.03	As relates to the Cyp11A1, in the discussion it is suggested to acknowledge that in addition to serving as a rate limiting enzyme of steroidogenesis it can activate, vitamin D3, lumisterol and tachysterol to biologically active metabolites as briefly reviewed in (Journal of Investigative Dermatology. 2023, 143(12): 2340-2342. doi: https://doi.org/10.1016/j.jid.2023.07.003). In this context local expression of this enzyme could play a positive homeostatic role.	Thank you for tyour insightful suggestion to acknowledge the broader functional role of Cyp11A1. We have expanded our discussion to include the potential positive homeostatic role of local expression of Cyp11A1, emphasizing its diverse and significant impact within biological systems. Newly added text in the discussion part: “CYP11A1 is the first and rate-limiting enzyme of the steroidogenesis pathway. However, recent studies suggest that it can activate vitamin D3, lumisterol and tachysterol to biologically active metabolites. Therefore, local expression of this enzyme in TNBC could play a homeostatic role by producing these metabolites.”
1.04	Again, the subject of local steroidogenesis within tumor deserves more consideration in a wider manner because of its clinical implications and implications for natural history of cancer. In this context mentioning that tumor can affect its environment and body homeostasis when advanced as discussed recently (How cancer hijacks the body’s homeostasis through the neuroendocrine system, Trends Neurosci 46: 263-275, 2023) would be appreciated by the readers.	We thank the reviewer for highlighting the importance of local steroidogenesis within tumours and its broader clinical implications, as well as its impact on the natural history of cancer. In response, we have expanded our discussion section to include an analysis of how cancer can manipulate body homeostasis through the neuroendocrine system, citing recent findings from "How cancer hijacks the body’s homeostasis through the neuroendocrine system" (Trends Neurosci 46: 263-275, 2023). Newly added text: Cancer not only escapes the body's regulatory mechanisms, but it also modulates local and systemic homeostasis, favouring tumour growth. Tumours produce cytokines, immune mediators, classical neurotransmitters, hypothalamic and pituitary hormones, many

	In summary, this is an interesting study that should be of strong interest to the readers of Nature Communications, after minor revisions as suggested above.	metabolites, and glucocorticoids to control neuroendocrine regulation of homeostasis as demonstrated in human and animal cancer models. Cyp11a1 activity, therefore, can mediate bidirectional communication between local autonomic and sensory nerves, and the tumour, with putative effects on the brain, is also envisioned. In this study we cannot exclude the possibility of posaconazole to inhibit such metabolite production beyond its ability inhibit steroidogenesis. Thank you.
--	--	---

Reviewer 2

2.01	The major aim of this study, entitled “Perturbing local steroidogenesis to improve breast cancer immunity”, was to evaluate the role of steroids in TNBC progression, exploring the possibility that steroids accelerate tumor growth by remodeling the immunosuppressive activity of the tumor microenvironment (TME). To this end, the authors combined multi-omics analysis (metabolomics, transcriptomic) to characterize steroidogenesis in TNBC tumors, and subsequently, conduct genetic and pharmacological perturbation studies, using conditional knockout of Cyp11a1, the rate-limiting enzyme of steroidogenesis, and the newly identified steroidogenesis inhibitor Posaconazole. The authors then convincingly showed in animal models that these perturbations significantly reduced tumor growth and decreased its volume. Using several functional assays and transcriptomic profiling the authors showed that steroidogenesis blockades affect immunosuppressive of tumor associated macrophages (TAMs) and enhanced anti-tumor immune response of dendritic and T cells. Based on further analysis (scRNAseq, tumour steroidomics) the authors suggest that mast cells	We sincerely thank you for dedicating time to review our manuscript and for your helpful suggestions. We found all the questions raised by the reviewer to be important and constructive.
------	---	--

	and basophils are the predominant steroidogenic agents in TNBC for enhanced therapeutic precision.	
2.02	Overall, it is a comprehensive study involved in multiple omics data and animal models, describing significant and promising effects on tumor growth. While these parts are convincing, the others are confusing and primarily descriptive, lacking a clear mechanistic data of how inhibition of steroidogenesis indeed attenuates tumor growth. There are many correlations and supporting evidence but not direct experimental data of mechanisms. Dealing with macrophage, T cells, dendritic cells, mast cells and basophils; it is unclear how all these works together and how to dissect the contribution of each for the end point outcome on tumor growth.	Thank you for your comprehensive feedback on our study, and we acknowledge your concerns about parts of the manuscript. In response to your comments, we have thoroughly revised the manuscript to clarify the underlying mechanisms more explicitly. We provide additional experimental data (see summary list of newly added Figs and their conclusions) to strengthen the link between steroidogenesis inhibition and tumour suppression. Specifically, we have elaborated on how reducing local steroid concentrations can lead to an activation of various anti-tumour immune cells, including macrophages, T cells, and dendritic cells. The revised sections now offer a clearer explanation of how these immune components interact and contribute to the observed effects on tumour growth. Briefly, tumoricidal state and tumour promoting state of these effector immune cell types are well known. The general principles of anti-tumour immunity and immune suppression are also well studied. Type-2 macrophages (M2) are known to be tumour promoting, while type 1 macrophages (M1) are known to be tumoricidal. While effector T cells are anti-tumoural, however, these effector T cells become exhausted and become dysfunctional in the immunosuppressive tumour microenvironment. The predictor gene expression pattern in these contrasting states are known. Dendritic cells that are able to present tumour antigens are tumoricidal by eliciting effective adaptive immune responses, however, they are defective and tolerogenic and DCs do the opposite in the TME. Their signatory gene expression pattern is also known. In this study we show that the presence of steroid, actively contributed by the steroid-producing immune cells, induce suppressive signature in all these effector immune cells. Steroid-producer cells produce steroids that diffuses freely and act on all immune cells in an autocrine and paracrine manner. All effector immune cells (T cells, macrophage, DC etc) are steroid responder cells as revealed by their glucocorticoids receptor (NR3C1) expression. The mechanism of steroid mediated induction of immunosuppression in several immune cell type is also known. The major finding here in this study is to show evidence and demonstrate such anti-tumour immune suppression induced by TNBC patients and mice model of TNBC. The role of mast cell and basophils in tumour killing is context dependent. Mast cells are reported to promote or restrict tumour growth depending upon context. In our study mast cells were found to be initiator of the steroid biosynthesis as they express CYP11A1. The contribution of each cell type is additive.

2.03	Specific points (1) Figure 1: Functional Steroidogenesis. (1A) The authors performed proteomics of 16 tumors from TNBC patients to characterize their steroidogenesis. The figure should include information of their molecular subtyping, at least if they are LAR (luminal androgen receptor) positive.	We thank you for this suggestion. The molecular subtyping of all 16 TNBC patients, including standard staining results for ER, PR, HER2, and Ki67, has been detailed in Supplementary Table S1.
2.04	(1B) It is unclear whether the profiling described in the figure is characteristic to TNBC as it was performed for only 16 samples. There are many studies describing metabolic data of TNBC patients, using much larger cohorts. For example- 10.1038/s41422-022-00614-0. Perhaps the authors can gain supportive evidence from larger cohorts of previous studies.	Thank you for your suggestion; we appreciate your input. However, after thorough consideration, we found that there is no previously published research available that specifically measures steroid levels in tumours. The referenced study you mentioned lacks direct steroid detection data, focusing instead on untargeted metabolomics, which is not applicable or informative for our specific research goals. Our method, utilising targeted LC-MS/MS, is better suited for this purpose, allowing us to accurately measure and analyse steroid levels in TNBC tumour tissues. This mentioned report was able to detect many metabolites but failed to detect any steroids or steroid derivatives (except pregnenolone sulfate) because steroid and sterol detection needs optimised steroid/sterol extraction and chromatographic/mass-spectrometric detection. The molecular basis for this is that the steroid molecules are hydrophobic. In our study biological samples were processed in a customised approach dedicated to detect the steroids and their derivatives.
2.05	(1C) The data is shown on TNBC. Is it specific for TNBC? what about non-TNBC?	This is specific to TNBC. We excluded non-TNBC because they respond to steroid hormones estrogen and progesterone. This is to avoid any redundant effect, if any. We wanted to focus on glucocorticoid signalling which was predicted from the scRNAseq analysis. Steroid hormone receptor expression in non-TNBC tumours is now presented in Fig S1F. Fig S1F

F. Bar plot illustrating the expression levels of various steroid hormone receptors in tumours from patients with breast invasive carcinoma, analysed using the GEPIA tool to access TCGA datasets.

2.06 (1F) Combining IF analysis to the IH data to show by co-staining the identity of CYP11A1-positive cells, at least with immune marker (CD45 positive).

Co-staining data is now provided (Fig 1H)

The new figure would appear as:

H. Representative immunofluorescence images of tumour sections of TNBC patients showing CD45 (green), CYP11A1 (red), and overlap.

2.07 (S1D) It will be more convincing to show FACS analysis of M2 macrophage markers and NR3C1.

We agree that it is a good suggestion. However, in this report, we would prefer to rely on gene expression data for NR3C1 and M2 marker gene expression. It is well established that glucocorticoid action via glucocorticoids receptor, NR3C1, induce M2 phenotype in

		macrophages. M2c name was coined to denote their glucocorticoids induced M2 phenotype (Martinez and Gordon et al., 2014. doi: 10.12703/P6-13). Available human samples are precious and ethically regulated, so we used them in more important/informative experiments.
2.08	(2) Figure 2: Deletion of Cyp11a1 in Immune Cells Restricts TNBC Tumour Growth (2A) Cyp11a1 KO should be shown by genetic analysis and by WB	Genotyping has been done by Transnetyx confirming the presence of Cre and LoxP allele. We have shown the knockout efficiency by western blot (Fig S2A) in the revised manuscript. It will appear as A  A. Western blot analysis of Cyp11a1 protein expression in splenocytes from control and Cyp11a1cKO mice. The upper panel shows the specific immunoreactive bands for Cyp11a1, and the lower panel displays β-actin as a loading control to ensure equal protein loading across the samples.
2.09	(2B) EO771.LMB implanted subcutaneously-orthotopic model would possibly be more suitable. EO771.LMB is metastatic clone of EO771, very different from EO771, which is usually used as a model for primary tumor- why this cell line was selected?	When possible, we did both orthotopic and subcutaneous model. Some experiments were done orthotopically (Fig. 5D, G-M) some were subcutaneous (Fig 2,3, 4 and 5A-C, E-F). Pengfei Qiu's lab has this expertise. Mahata lab relies on subcutaneous. We used both the lines. Fig 5D was with EO771 and orthotopic. There was no specific reason to use EO771.LMB. It was available immediately and represents metastatic breast cancer. In the literature, we found both the lines were used in subcutaneous/orthotopic syngeneic model and accepted as model to test immune responses (e.g.,DOI: 10.1038/s41467-019-12222-5).
2.10	(2F) The results showed significant upregulation of NK cells- any impact?	NK cells are known for their tumour killing activity. They respond to glucocorticoids and glucocorticoids suppress NK cell function. This could be the one of the reason why we observed tumour restriction due to the pan-hematopoietic Cyp11a1 knockout. A dedicated research article on the impact of NK cells, as a steroid responder cells, in anti-tumour immunity (in lung cancer) is under preparation. If needed, we will be happy to share the manuscript. We have discussed the result briefly elsewhere in the revised manuscript altogether with other effector immune cells.

2.11	(2i) MertK was used as a marker for M2; other canonical markers CD206/163/FIZZ1 are usually used.	Mertk is also surrogate of M2. We have now analysed Arg1, iNOS and presented in Fig 2K, L. Fig 2K, L  K. Flow cytometry visualisations indicating iNos expression in CD11b+ cells of Vav1Cre versus Cyp11a1cKO mice (left). The subsequent chart (right) illustrates the comparative percentage expression in both mouse types (N = 3 ; Mean ± SEM expressed through error bars). L. Flow cytometry visualizations indicating Arg1 expression in CD11b+ cells of Vav1Cre versus Cyp11a1cKO mice (left). The subsequent chart (right) illustrates the comparative percentage expression in both mouse types (N = 5; Mean ± SEM expressed through error bars).
2.12	(3) Figure 3: Steroidogenesis Inhibition Augments Anti-Tumour Immunity (D-I) in vitro analysis of monocytes isolated from peripheral blood ± GC treatment highlights possible role of immunoregulatory cytokines such as IL-10, TGFb1 etc that might have impact on the tumor growth. Will be important to look on monocyte (DC) from the tumors (control and KO) and evaluate the expression levels of these cytokines.	In DCs, pan-hematopoietic Cyp11a1 knockout (Cyp11a1^{ckO}) decreases the tolerogenic DC score (Fig S3 H) and increases the DC maturity score (Fig S3 I). Tolerogenic DC score suggest the ability of inducing tolerance. DC maturity score suggest their antigen presenting ability. This approach we believe more reliable than depending on one or two gene expression. Now, in the revised manuscript, we checked IL10 and TGFβ expression in DCs as suggested by you and found Cyp11a1 knockout decreases the IL10 and TGFβ expression in DCs (Fig 3 K and Fig S3 J respectively). Fig 3K and Fig S3 H, I, J will appear as: Fig 3K

K
K. FACS histogram on the left, displaying the expression intensity of IL-10 in DCs from Vav1-Cre (control) or Cyp11a1^{CKO} mice. The complementary right panel juxtaposes percentage expression in both cohorts (N = 5; Mean ± SEM demarcated by error bars).

Fig S3 H-J**H****I****J**
H-I. Violin plots reveal the relative significance of indicated gene set signature scores in DCs and T cells, drawn from either Vav1Cre or Cyp11a1cKO mice.

J. FACS histogram on the left, displaying the expression intensity of TGF-β in DCs from Vav1Cre or Cyp11a1cKO mice. The complementary right panel juxtaposes percentage expression in both cohorts (N = 5; Mean ± SEM demarcated by error bars).

Will be also important to define the ratio between M1 and M2 macrophages from in vivo tumors.

We have now analysed Arg1, iNOS (in addition to the previous Mertk expression) and presented in Fig 2K, L.

Results (Fig 2K, L) are:

K. Flow cytometry visualisations indicating iNos expression in CD11b+ cells of Vav1Cre versus Cyp11a1cKO mice (left). The subsequent chart (right) illustrates the comparative percentage expression in both mouse types (N = 3 ; Mean ± SEM expressed through error bars).

L. Flow cytometry visualizations indicating Arg1 expression in CD11b+ cells of Vav1Cre versus Cyp11a1cKO mice (left). The subsequent chart (right) illustrates the comparative percentage expression in both mouse types (N = 5; Mean ± SEM expressed through error bars).

2.13 (4) Figure 4: Characterization of Cyp11a1+ Immunocytes in TNBC

The author suggest that mast cells are the predominant population that “modulates immune response in TNBC by contributing to the local immunosuppressive steroid biosynthesis (Figure 4D, 4F). The immune suppressive capacity of mast cells is attributed by the secretion of IL-10, TGFβ, prostaglandin D2’s that skew dendritic cell function driving a Th2 immune response”.

While this is a feasible possibility, the data presented in figure 4 doesn't prove it, only provides some correlative findings, including data shown in Fig. 4G, which could be not related to the presented study. Demonstrating the role of

We agree with you that it is a speculation. We have rephrased the claim (by rewriting that paragraph in the discussion section) to bring more clarity and keep conclusions evidence-based. Speculations are now clearly mentioned.

The discussed paragraph will appear as:

“Observation of Cyp11a1-mCherry+ myeloid cells (macrophages/monocytes) in the tumour-bearing Cyp11a1-mCherry+ mice was consistent with the previous observation of colorectal cancer by Acharya et al., *Immunity*, 2020; but dissimilar with melanoma by Mahata et al., *Nat Commun*, 2020, where the percentage of Cyp11a1+ macrophages/monocytes were very low by number. This observation indicates the heterogeneity of the tumour microenvironment. Induction of immune cell steroidogenesis in immune cell types seems to be tumour-type and mice model-specific. Observation of mast cell expression of Cyp11a1 is new. Mast cells, traditionally perceived as key players in allergic responses and inflammatory processes, have in recent years, gained recognition for their multifaceted roles within the tumour development.

	mast cells as key mediators of steroidogenesis associated TNBC progression is required.	Previous reports claim increased mast cell presence with enhanced breast cancer angiogenesis 50,51. Our study suggests that mast cells modulate immune responses in the TNBC TME by contributing to the local immunosuppressive steroid biosynthesis (Figure 4D, 4F). The immune suppressive capacity of mast cells is attributed by the secretion of IL-10, TGFβ, prostaglandin D2's that skew dendritic cell function driving a Th2 immune response61,62. Our identification of Cyp11a1+ steroidogenic mast cells within the TNBC TME posits them as instrumental in producing immunomodulatory steroids, aiding the immune escape tactics of cancer cells, possibly by inducing tolerance in DCs. However, further experimentation is needed for direct evidence. Functionally similar basophil's role in this study is not clear, particularly in humans."
2.14	(5) Figure 5: Posaconazole Diminishes TNBC Progression Characterizing the immune landscape of control and Posaconazole-treated tumors is required, considering all the above findings, including M1/M2 macrophage ratio, mast cells etc.	To address this question, we compared transcriptomes of posaconazole treated and untreated tumours (syngeneic orthotopic E0771 tumour in the breast fat pad) and displayed a global views of differential gene expression using Cibersort analysis (Fig 5D). We observed an improved immune response, such as increase in activated T cells, M1 macrophages, and decreased tolerogenic DCs as expected. Fig 5D  D. Heatmap depicting the immune cell populations quantified by Cibersort in tumors from Posaconazole-treated and control mice in a TNBC mouse model (E0771 injected orthotopically).
2.15	(6) A model/scheme describing the major findings could help	Thank you for the suggestion. A conceptual diagrammatic summary is now provided.

Reviewer 3

3.01 Immunotherapy in breast cancer has limited efficacy, with the most promising response being observed in triple negative breast cancer (TNBC), a highly aggressive form of the disease. Identifying mechanisms of immune suppression in TNBC should provide avenues for improving response to immunotherapies. Given the ability of glucocorticoids to suppress immune function, this study aims to identify whether local steroid hormone production may impact infiltrating immune cells in TNBC. Using LC-MS/MS analysis of human TNBC samples, they identify a set of steroids that are present in tumors. Using a larger cohort of patient samples, they further demonstrate that immune cells express the mRNAs for a variety of steroid receptors,

We sincerely thank you for dedicating time to review our manuscript and for their helpful suggestions. We found all the questions raised by the reviewer to be important and constructive.

	suggesting that immune cells could respond to steroids in the local microenvironment. Immunostaining for the rate limiting step of steroid biosynthesis, CYP11a1/p450scc, revealed high expression in the immune component of tumors and that immune cells in the tumor have a high steroid biosynthesis score. Thus, the authors ask whether steroid production by immune cells can impact tumor immunity and if this activity can serve as a therapeutic target. Supporting this concept, targeted disruption of Cyp11a1, which will reduce steroid production, in immune cells suppresses TNBC growth. It also shifts the immune composition of tumors. Direct treatment of dendritic cells in vitro with cortisol shifted their cytokine profile. Single cell RNA-seq data further suggested that basophils and mast cells have increased likelihood for producing pregnenolone, the derivative of cholesterol produced by CYP11A1. Evaluating patient data demonstrated that increased mast cell number was associated with worse patient outcomes. Lastly, to demonstrate the importance of CYP11A1 in tumors, a humanized mouse model of TNBC was treated with posaconazole, an inhibitor of CYP11A1, and the growth of tumors was suppressed and this was associated with changes in the expression of cytokine signature gene sets.	
3.02	While immune cell production of steroids is an intriguing regulator of tumor immunity. There are several major weaknesses with the current study that limit its impact. 1) There is limited novelty in the study. A) The senior author published a Nature Comm. study in 2020 indicating that steroid biosynthesis in T-cells was important for tumors to suppress immunity using many of the same approaches, but in a melanoma model.	Thank you for your comprehensive feedback on our study, and we acknowledge your concerns. A) We agree that the present study is a continuation of the previous report (Mahata et al., Nat Commun, 2020) on local steroid biosynthesis by immune cells. The previous study was further supported by Nandini Acharya et al., Immunity, 2020. Those two studies raised the possibility of context dependent maladaptation of the immune cell steroidogenesis and steroid signalling in the tumour; a proposed mechanism of immunosuppression and immune evasion of tumour. Human relevance was predicted but not proven by direct steroid analysis.

	B) The authors also report that posaconazole is a “newly identified drug” that inhibits steroidogenesis. This drug was FDA approved in 2006 and reported to be an inhibitor of CYP11A1 in 2013 (Mast, et al. Mol. Cell. Endocrinol.). It is highly related to ketoconazole, which has been used for decades to block steroid production. It has been shown to inhibit the growth of many tumor types. C) It is well established that glucocorticoids will impact dendritic cells in vitro. The analysis shown in the current manuscript is not particularly informative. D) In contrast to the TIMER RNA-based analysis shown in this study, mast cells have been examined in a tissue microarray of a large cohort breast cancer patients (4,444 cases) and	Novelty of this study includes relevance to TNBC with human evidence: (1) Presentation of steroid level in human tumours has not been published previously. We show quantitative steroid profiling by mass-spec. (2) Demonstration of mast cells as steroid producers in TNBC is new. (3) Posaconazole as a therapeutic option for TNBC with human data along with in vivo mice data along with humanised mice model. (4) Dendritic cells are revealed as steroid-responder cells. Steroid-induced tolerance in DCs explains why regulatory T cells are more in tumour and reduced in pan-hematopoietic Cyp11a1 knockout (Cyp11a1^{ckO}) mice. (5) It contains large amount of steroidomics and transcriptomics data that can be used as resource in the future. B) The reviewer is correct and we have now rephrased it to make our claim clear. The use of posaconazole as a therapeutic strategy in breast cancer to improve cancer immunity by suppressing steroidogenesis and steroid signalling is novel. We identified posaconazole as a drug via drug repurposing study. A patent application is pending. By saying “newly identified drug” we mean it is a new drug for the treatment for TNBC and maybe other steroidogenic tumours. We rephrased it in the revised manuscript to make it clear. C) Surprising but true that no genome-wide human DC’s transcriptomics changes because of endogenous glucocorticoid (cortisol) treatment (i.e., RNAseq data on cortisol treated DC) is available up to our knowledge. We searched for such resource to build our hypothesis but did not find. Reports on candidate gene expression analysis is also limited for endogenous glucocorticoid (cortisol) and limited to a few number of genes. Most inferences are based on synthetic glucocorticoids (e.g., dexamethasone). Therefore, it was worth to do this in vitro RNAseq analysis and subsequent flow cytometric or ELISA validation experiments to reveal the genome-wide effect of endogenous glucocorticoid cortisol. This gene expression data set will be a rich resource to be used by the future researchers to check any gene of interest. However, the results were expected. Immunosuppression by glucocorticoids is well established. Therefore, it is obvious to hypothesise that if glucocorticoids are in detectable and functional level, then it will suppress anti-tumour immunity by inducing tolerance in DCs. Therefore, the presence of immunosuppressive steroid glucocorticoids and its cognate receptor expression is an important and novel finding. D) Thank you for bringing up the discrepancy regarding the TIMER RNA-based analysis in our study and the mast cell (MC) association with good outcomes in a tissue microarray (TMA) of a large cohort of breast cancer patients (DOI: 10.1007/s10549-007-9546-3). Below are explanations to address this discrepancy.
--	---	--

associated with good outcomes. This is a major discrepancy and that the authors should address.

Patient Cohort Specificity: The patients included in the study you referenced were all early-stage breast cancer patients (as outlined in the Materials and Methods section under Patient Selection). The characteristics of the patient cohort in terms of cancer stage and lymph node involvement can heavily influence outcomes, which may contribute to the observed discrepancies.

Breast Cancer Subtype Differences: In the TMA cohort, ER-positive cases accounted for 78.1% (as shown in Table 2), which could be a key factor explaining the different findings regarding MCs.

Limitations of TMA: Tissue microarrays (TMAs) have limitations, particularly in representing the entire tumour's microenvironment. TMAs typically sample a small portion of the tumour, which may not capture the full heterogeneity of mast cell distribution and characteristics within the tumour. This limitation could contribute to the differing conclusions drawn between TMA-based studies and more comprehensive analyses like ours.

Different Survival Endpoints: The study you referenced analysed breast cancer-specific survival (BCSS), whereas our study focused on overall survival (OS). The choice of survival endpoint could introduce variability in the association between MCs and patient outcomes. Disease-specific survival curves, also known as cause-specific survival curves, measure the percentage of people in a treatment or study group who have not died from a specific disease within a defined period of time. The time period usually starts at the time of diagnosis or treatment and ends at the time of death. The curve uses death from the disease of interest as the endpoint, so it doesn't include patients who die from other causes or relapse. It also doesn't always include deaths caused by disease-related factors, such as treatments. Because of this, disease-specific survival curves can be misleading because they are always higher than overall survival and disease-free survival curves. (10.1016/j.otohns.2010.05.007). Therefore, we used cumulative overall survival curve. We used TIMER available transcriptome-based analysis because it not just includes large cohort but also takes account of a whole battery of known genes.

cKit is not the only determining factor to infer mast cell: The inference in the reviewer-mentioned reference study (doi: 10.1007/s10549-007-9546-3) has been made based on tissue microarray data on cKit expression. cKit expresses in many other immune cells (NK cells, T cells and dendritic cells) as well as bone-marrow-mobilised haematopoietic stem cells and progenitor cells. Therefore, the conclusion may be erroneous.

Finally, in that study, they showed that the presence of even one (or more) MC in the tumour microenvironment was sufficient to exert a positive prognostic effect. They did not find a statistically significant prognostic effects of MCs in the node-negative group in the training set analysis. This differs from their earlier published report (doi:10.1038/modpathol.3800094) on

a 348 case series which concluded that the presence of MCs was a favourable prognostic factor in the node-negative patients ($P = 0.018$) but not the node-positive group ($P = 0.384$).

Mast cells have been found with anti-tumour as well as tumour promoting role in different tumour types depending up on the context. Further research is needed to understand the molecular reasons behind these contradictory observations.

This has been now discussed in the “discussion section”. However, we are not comfortable to criticise their study, rather to strengthen our conclusion, we added more analysis on mast cell and Cyp11a1 expression in patient survival (Fig 4G, bottom panel). In this additional analysis revealed high mast cell infiltration in combination with high Cyp11a1 expression brings worst survival outcome.

Fig 4G (bottom panel)

G. Kaplan-Meier curve elucidating the correlation between both mast cell infiltration (up) and the combined impact of CYP11A1 mRNA expression with mast cell infiltration (bottom) on survival outcomes in breast cancer (BRCA) patients, analysed using the TIMER database.

3.03 2) There is no consideration of whether immune cell production of steroids has any impact on local concentrations of steroids. If not, the authors can only conclude that CYP11A1 function is important for immune cell function, but not that steroids are involved.

We thank the reviewer for this suggestion.

We have now steroid profiling data on tumours (Fig 2D) and included in the revised manuscript. We found that local concentration of steroids in the tumour decreased when Cyp11a1 was deleted in all immune cells using Cyp11a1^{fl/fl};Vav1-cre (Cyp11a1^{CKO}) mice, suggesting the active de novo synthesis by immune cells.

Fig 2D

D. Bar chart showing the scaled concentrations of steroid hormones detected in the tumours from control and Cyp11a1^{CKO} mice via LC-MS after organic solvent extraction (N = 6 for control and N = 9 for Cyp11a1^{CKO}).

Do tumor cells also produce steroids? More importantly, are the steroids that are produced by the ovary and adrenal gland, and certainly in the local environment of the tumor, higher than what is produced by the immune cells? These issues should be addressed because it is not clear whether inhibition/knock-out of CYP11A1 cause changes other than steroid levels, such as accumulation of cholesterol that could shunt this key metabolite to other pathways, ultimately impacting immune cells. Direct assessment of local steroid levels in the immune KO of Cyp11a1

No, tumour cells (E0771.LMB) do not produce any steroids (Fig S2B). There are no changes (or insignificant changes) in the systemic level of steroids (Fig S2C) as revealed from the steroid profiling of the tumour bearing serum of Cyp11a1^{CKO} and control mice.

Fig S2B, C

is needed to demonstrate a steroid-based immunomodulatory effect.

B. Bar chart showing the scaled concentrations of steroid hormones detected in the E0771.LMB cells and tumours from control mice via LC-MS after organic solvent extraction (N = 3 for cancer cells and N = 6 for tumour tissues).

C. Bar chart showing the scaled concentrations of steroid hormones detected in the serum samples from control and Cyp11a1cKO mice via LC-MS after organic solvent extraction (N = 10).

Maybe topical to discuss that tumour cells may acquire the capacity of steroid biosynthesis or local steroid conversion in a context dependent manner. For example, in prostate cancer, cancer cells by themselves acquire de novo steroidogenic capacity (DOI: 10.1158/0008-5472.CAN-07-5997).

3.04 3) The systemic drug treatments cannot be used to make conclusions regarding immune cell production of steroids because these treatments will change circulating levels of the same steroids.

We agree with the reviewer. At present there is no available immune cell specific delivery system that we can use. However, it is expected that posaconazole would stop immune cell steroidogenesis (which is weaker and effective in autocrine and paracrine manner locally) more effectively compared to glandular steroidogenesis (which is robust and for systemic supply of steroids effective in endocrine manner). In vitro, posaconazole inhibits Cyp11a1 activity both in adrenal gland-derived cells and immune cells. However, in vivo, in mice, we did not observe any significant change in steroid level. A research paper (Pramanik et al., 2024, iScience, under review) and respective patent application is under review. We will be happy to share if needed. However, very higher dose of Posaconazole is expected to block all glandular steroidogenesis.

3.05 4) The authors focus on glucocorticoids to the exclusion of androgens and progestins based on an RNA analysis. They should directly assess the impact of these other steroids to warrant conclusions regarding glucocorticoids. In addition, there is a failure to discuss the highly controversial roles of glucocorticoids in breast cancer that have been previously reported. In some cases, glucocorticoids have been reported

We hypothesised glucocorticoids because of the target receptor and associated target gene expression. As we didn't observe any evidence to suspect ER or progesterone in this context, so we did not follow up. Potential role of ER and PR are now discussed.

The paradoxical role of glucocorticoids is now discussed in the introduction section.

The newly added text will appear as:

“The majority of breast cancers respond positively (steroid sensitive) to the sex steroids, estrogen and progesterone, because most breast cancer types are either ER and/or PR

	to be tumor promoting and in other cases, tumor suppressing in breast cancer.	positive. Estrogen and/or progesterone signaling promote their survival and growth. TNBC do not respond to sex steroids, but they do respond to glucocorticoids. Glucocorticoid signaling in breast cancer has a complex role and can be paradoxical depending on context. Overall, they promote TNBC by various means. GR activation can stimulate proliferation, help cells escape apoptosis, decrease cell adhesion and stimulate cell motility, which can increase the risk of metastasis. High GR expression in early-stage TNBC is associated with chemotherapy resistance and increased recurrence. GR expression can be a prognostic biomarker for TNBC. Therefore, GR antagonists, such as mifepristone, may be used in conjunction with chemotherapy to treat TNBC. Yet, the potential of de novo steroidogenesis in the TME and GR signalling to accelerate TNBC progression by sculpting an immunosuppressive microenvironment remains an uncharted territory.”
3.06	Minor comments: 1) The mast cell survival analysis data are unrelated to local production of steroids. Thus, there contribution to the study is marginal.	The purpose of displaying mast cell survival data is to show that the inhibition was not due to the non-specific induction of cell death.
3.07	2) The treatment protocol for posaconazole is not included. What was the dose, the dosing schedule, and the route? Most importantly, what was the size of tumors prior to treatment or was treatment initiated when tumor cells were implanted? If the drug was given at the time of tumor cell implantation, this has no relevance to human disease.	We apologise for the mistake that treatment protocol was missing from the materials and methods section. We have now included this in the revised manuscript and appeared as follows “Posaconazole treatment Mice received Posaconazole (Noxafil, 40mg/ml oral suspension) by oral gavage at 20 mg/kg body weight, diluted in water, once daily. In some experiments (subcutaneous), the first dose was administered concurrently with cell injection and continued for 10 days. In others (orthotopic), we administered the first treatment 7 days after the tumours were established. The vehicles were administered orally by gavage in control mice.” The dose was determined by the previous research reports (e.g., DOI: 10.1093/jac/dks090)) and after discussion with our University’s veterinary surgeons who oversee our animal work. We choose lower dose to avoid adrenal insufficiency due to systemic blockade of glandular steroidogenesis. In mice experiments, Posaconazole is generally used between 5 mg to 100 mg/kg body weight. The experiments are designed to test the effect of drug to prove the concept/principle administering concurrently with cancer cell injection, and also, after tumour was established (7 days post cancer cell injection). Human relevance will be revealed after completion of the drug trial which is under way. We are initiating a multicentre Phase II clinical trial. The trial has already received approval from the ethics committee at the lead centre (Ethics Approval No.: SDTHEC2024002011), and we are currently in the process of registering the clinical trial.

3.08	3) There is inconsistency about the site of the xenografts. In some cases “subcutaneous” is stated and in others “orthotopic”. Were the tumor cells injected directly into the mammary fat pad?	Both are correct. Some experiments were done orthotopically (mammary fat pad injection) (Fig. 5D, G-M) some were s.c. (Fig 2,3, 4 and 5A-C, E-F). Pengfei Qiu’s lab has this expertise. Mahata lab relies on subcutaneous.
3.09	4) Posaconazole has also been reported to inhibit CYP46A1.	We have now discussed on off-target effect of posaconazole elsewhere in the revised manuscript.

Reviewer 4

4.01	Introduction - Would benefit from more explanation of what steroid hormones are, and whether you are referring to corticosteroids or sex steroids (like in the Mahata Nature Comms paper https://pubmed.ncbi.nlm.nih.gov/32680985/)	We sincerely thank you for your generous appreciation of our work and for taking the time to provide valuable suggestions. Thank you for this suggestion. This has been done in the revised manuscript (throughout).
4.02	- Line 61-62 – would not say ‘limited treatment avenues’; there are many treatments available, however they are not targeted options and have suboptimal clinical efficacy. Recommend more recent citations here – a lot has changed since 2010.	Thank you for the suggestion. We have updated the introduction as suggested and updated the references with recent publications. That sentence will appear as “Among its subtypes, triple-negative breast cancer (TNBC) emerges as a particularly aggressive type, and lacks specific targeted and effective therapy ”.
4.03	- Line 73 – “principal drivers” might be a little strong – macrophages are important but not the ‘principle’ driver of tumour growth and metastasis	We have toned down it to “ one of the major drivers ” in the revised manuscript.
4.04	- Line 92 – I think “responsive to steroid hormones” needs to be explained further – does responsive mean encourages or discourages growth? Also I think it would be useful to say corticosteroids here, TNBC is by definition not responsive to sex steroids	We have now revised the introduction and clarified as suggested.
4.05	- Line 114 - What does ‘analytic experimental assays’ mean?	We have now rephrased it: “ in vitro assays and in vivo experiments in mice models ”
4.06	- Line 124 – posaconazole is not newly identified – it’s an antifungal that’s been used clinically for almost 20 years. Do you mean newly identified in use for inhibiting steroidogenesis?	We are sorry for the confusion made. We have now clarified this by revising the text. Yes, we mean it is newly identified drug (as revealed by a structure-based drug repurposing study) to be tested in steroidogenic tumours to inhibit Cyp11a1 activity.
4.07	Results	

Functional Steroidogenesis and Steroid-Signalling Exist in TNBC Tumours

- scRNA-seq has shown steroidogenesis signature in immune cells compared to peripheral blood, but it is possible to tell if the tumour cells themselves are also synthesising steroid?

Similar to point 3.03.

We don't have any evidence supporting cancer cell-mediated steroidogenesis by TNBC. One such representative human *CYP11A1* expression data in TNBC cancer cells is now displayed in Fig S1 G, which so no expression of it.

Fig S1G

G. Violin plot depicting the expression level of *CYP11A1* within epithelial cells from 5 TNBC patients.

In mice also, tumour cells (E0771.LMB) do not produce any steroids (Fig S2B). There are no changes (or insignificant changes) in the systemic level of steroids (Fig S2C) as revealed from the steroid profiling of the tumour bearing serum of *Cyp11a1^{ckO}* and control mice.

Fig S2B, C

B. Bar chart showing the scaled concentrations of steroid hormones detected in the E0771.LMB cells and tumours from control mice via LC-MS after organic solvent extraction (N = 3 for cancer cells and N = 6 for tumour tissues).

C. Bar chart showing the scaled concentrations of steroid hormones detected in the serum samples from control and Cyp11a1cKO mice via LC-MS after organic solvent extraction (N = 10).

4.08 Genetic Deletion of Cyp11a1 in Immune Cells Restricts TNBC Tumour Growth and Alters Immune Infiltration in the TME
 - This Cyp11a1 null mouse model is a nice experiment.
 You show that there are differences in the immune infiltrate in tumours, but is there a difference in the systemic immune profile of cells? Is it possible that the Cyp11a1 is influencing the production or maturation of immune cells and this is why you see a shift?

We had the similar question in earlier study, too. Previously we did not see any difference in T cell development, maturation and systemic level distribution because of Cyp11a1 deletion (Mahata et al., Nature Communications, 2020). Previous reports (Mahata et al., Nat Commun, 2020; Acharya et al., Immunity, 2020) suggest no changes in the systemic level immune phenotype upon Cyp11a1 deletion. In tumour bearing mice no changes observed in the draining lymph node or systemic level. The changes were only observed in the tumour (Acharya et al, Immunity, 2020; Mahata et al., Nature Communications, 2020). In Acharya et al., Cyp11a1 was deleted in myeloid cells using LysM-cre and in Mahata et al., Cyp11a1 was deleted in T cells. Similar to previous studies, in this present study, we didn't see any difference in the systemic/peripheral distribution of effector immune cell such as T cells, NK cells and myeloid compartment (Fig S2J) and their effector gene expression that determines their tumoricidal activity (Fig S2 K-Q).Fig S2J

Fig S2 K-Q

On the other hand, I find this experiment extremely hard to interpret – how can it be that profoundly altering steroidogenesis across all immune types (which could be the same as completely disrupting immune function) inhibits tumor growth. Why do we not see an acceleration of tumor growth when ALL immune cells are impacted?

J. Comparative analysis of immune cell populations in the spleen samples from control and Cyp11a1^{CKO} mice. The graph shows the percentage of CD4+ T cells, CD8+ T cells, NK cells, myeloid cells, and dendritic cells in the spleen.
 K-Q. Bar illustrates the comparative percentage expression of Arg1, iNos, and CD206 in CD11b+ cells (K-M), IL-10 and TGF-β in DCs (N-O), and IFN-γ and TNF-α in CD8+ T cells from the spleen samples from control and Cyp11a1^{CKO} mice.

We have now revised the manuscript thoroughly to increase the clarity. Vav1-Cre express in all immune cells during their hematopoietic development. So, this genetic approach (Cyp11a1^{CKO}) restricts all possible immune cell-mediated steroid production. Steroid hormones freely diffuse in the tumour microenvironment and come in contact with all cells (autocrine and paracrine activity). The effect that we see as phenotype is their “responder activity”. This “responder” activity of immune cells is due to the expression of steroid receptor (i.e., glucocorticoids receptor). In wildtype (Cyp11a1 sufficient mice) anti-tumour immune cells become suppressed because of the steroids. In Cyp11a1 knockout mice, absence of steroids reinstates their anti-tumour immunity.

Also, we have now provided the following diagrammatic summary of the findings that may help understand.

What happens if you put wild type macrophages back into the CYP11Ako mice. Do the tumors grow again?

Macrophages are plastic cells. The outcome of the experiment would depend on what type of macrophages we transfer and how they respond to the tumour microenvironment. We expect if we put steroidogenic macrophages or any other steroidogenic immune cell types into the

	Did you measure neutrophils? I can't see them in figure 1E or F? I find this odd. Figure 2I-J – are there fewer total immune cells in these mice? This has not been properly quantitated.	Cyp11a1 knockout mice, tumour may grow again. It would be a series of complex experiment to come with a conclusion. No, unfortunately we did not measure neutrophils. We lose them from the analysis. Sorry for the confusion because of the displayed figures. Now we have presented equal number of cells in the display. We did not quantify absolute number of immune cell infiltration, but measured the percentage of each population within CD45 positive immune cell population.
4.09	- Critically, I don't understand this experiment in the context of the authors own previous work (Nat Comm 2020, showing that specifically inhibiting CYP11A in Tcells is enough to inhibit TNBC tumor growth? In that paper, the authors show that inhibiting CYP11A in Tcells is enough to curb TNBC tumor growth? This work contradicts the current findings of this paper suggesting that it is macrophage specific Cyp11A1 that is critical in TNBC. At the very least, those findings need to be properly discussed at the beginning of this paper.	We apologise for the confusion again. We agree with the reviewer that the introduction and discussion was not enough to clarify this. We have now explained/discussed those previous findings further detail in the introduction. The conclusion from this study is also explained further. This study does not contradict the previous study. In the previous study, we observed T cell specific deletion of Cyp11a1 restricts tumour growth in B16 (subcutaneous injection) and E0711 (i.v. injection for lung colonisation/metastasis). We did not perform subcutaneous model of E0771 in Cyp11a1 knockout mice. In this study, we decided to use Vav1-cre to delete Cyp11a1 in all immune cells to block all possible immune cell source of steroids. Therefore, the current study does not contradict but inform us more TNBC focussed facts, despite opening up the complex context dependent induction of immune cell steroidogenesis. We have now revised the introduction and discussed previous finding further.
4.10	- The application of exogenous glucocorticoid to DCs and the resulting immunosuppressive phenotype (Fig 3E-J) is not new.	Surprising but true that no genome-wide human DC's transcriptomics changes because of endogenous glucocorticoid (cortisol) treatment (i.e., RNAseq data on cortisol treated DC) is available up to our knowledge. We searched for such resource to build our hypothesis but did not find. Reports on candidate gene expression analysis is also limited for endogenous glucocorticoid (cortisol) and limited to a few numbers of genes. Most inferences are based on synthetic glucocorticoids (e.g., dexamethasone). Therefore, it was worth to do this RNAseq analysis and subsequent flow cytometric or ELISA validation experiments to reveal the genome-wide effect of endogenous glucocorticoid cortisol. This gene expression data set will be a rich resource to be used by the future researchers to check any gene of interest. Immunosuppression by glucocorticoids and induction of tolerance in Dcs is well established. Therefore, The results were expected, but the global gene expression view is new.

4.11	Characterization of Cyp11a1+ Immunocytes in TNBC for Enhanced Therapeutic Precision - The focus on mast cells is again odd. In previous work (see above) the authors suggest it is Tcells. In this paper it is now potentially mast cells, DCs or macrophages.	The additional emphasis on mast cell in this paper is mainly because this information is new and relevant to human. The main focus of the paper is to show that local steroids, particularly glucocorticoids signalling suppress TNBC immunity in human and mice. We observed differences in Cyp11a1-expressing cells, but this difference possibly because of the heterogeneity of tumour types and model system used. In the previous report we used B16 (both subcutaneous and lung colonisation) and E0771 in lung colonisation (i.v.). As the presence of Cyp11a1-expressing mast cells and basophils were also observed in the previous study (minor population compared to T cells, Mahata et al., Nat Commun, 2020), and in macrophages (by Acharya et al., Immunity, 2020 in MC38 colorectal cancer model), in this study we used Vav1-cre to delete Cyp11a1 in all immune cells to block all possible sources of immune cell steroids.
4.12	- Mast cells identified as responsible for steroidogenesis (insufficient evidence in human samples for basophils)	Agree. We have mentioned this in the discussion part (quoted from the revised manuscript: "...basophil's role in this study is not clear, particularly in humans").
4.13	- Line 273 – careful with wording, posaconazole is not new	Thank you for pointing this mistake. We apologise for the misunderstanding because of inappropriate use of "new". The use of posaconazole as a therapeutic strategy in breast cancer to improve cancer immunity by suppressing steroidogenesis and steroid signalling is new. We identified posaconazole as a drug via structure-based in silico drug repurposing study. A patent application is pending. By saying "newly identified drug" we mean it is a new drug for the treatment of TNBC and maybe other steroidogenic tumours. We rephrased it in the revised manuscript to make it clear.
4.14	Newly Identified Drug Posaconazole Diminishes TNBC Progression in Preclinical Models - Your results in Fig 5 are nice but I am not convinced that this is due to posaconazole inhibiting CYP11A1	An entire manuscript is under review in iScience (Pramanik et al.). Because of a pending patent application we are unable to upload it into Biorxiv and cite immediately. However, if needed we are happy to share both the patent application and manuscript "in confidence". We will refer to that paper when available (will upload on to Biorxiv soon). Also, we provided steroid profiling data from Cyp11a1 knockout mice. (Fig 2D, S2B, S2C, see point 3.03 for figs) Figure copied from Pramanik J et al., iScience (under revision) to show the inhibitory effect of posaconazole compared to another well-known Cyp11a1 inhibitor, aminoglutethimide (AMG). Pregnenolone was measured in steroidogenic murine Th2 cells. [REDACTED]

4.15	- Importantly, what are the effects of the posaconazole on Mast cells, DCs, macrophages as per the previous figures?? Why are Tcells now effected? These results are very confusing.	I think the confusion is because previously it was not clear distinguishing the Cyp11a1-expressing (steroid producer cells) from steroid-responder cells (steroid-receptor expressing cells that responds to steroids). Steroid hormones freely/randomly diffuse from the steroid-producer cells because of their membrane permeable nature due to hydrophobicity and available to all immune cells to respond whoever express cognate receptors. We have revised the manuscript to make the steroids' intracrine and paracrine effect these clear. Also, the conceptual summary figure (Fig 6) may help with this (see point 2.15 and 4.08). Briefly, T cells are always dysfunctional in normal tumours because of presence of immunosuppressive steroids (glucocorticoids). In the absence of steroids in Cyp11a1^{ckO} they are effective. T cells are steroid-responder cells in this context and not steroid-producer. They respond to the local steroids available in a paracrine manner within the TME.
4.16	- Broadly – posaconazole (and all azole antifungals) are dirty drugs with innumerable off-target effects. There is already a lot of research looking into repurposing antifungal drugs in cancer across an array of targets including metabolism, angiogenesis, stroma, signalling pathways like AKT and mTOR.	We agree with you. A nice recent in-depth review is also available on this: 10.1016/j.jare.2022.08.018 However, those questions will be resolved in the future drug trials as well as further studies.
4.17	- I'd encourage you to consider the safety of antifungal drugs in cancer patients (https://academic.oup.com/jac/article/65/3/410/745723)	Thank you for the advice. We agree. To date no drugs are free from their potential adverse effects and off-target effects. Further research needed for its safety and efficacy. We are initiating a multicentre Phase II clinical trial on posaconazole's use in TNBC (see point 4.20 below)
4.18	- This drug is known to lead to adrenal insufficiency in humans, requiring administration of dexamethasone, a GC agonist to mitigate!	Adrenal insufficiency is because of prolonged use and associated with high dose. As immune cell steroidogenesis is significantly weaker than adrenal steroidogenesis, therefore it is expected that the dose that will be sufficient to stop immune cell steroidogenesis would not be able to stop the adrenal steroidogenesis completely. Posaconazole has been tested in many experiments in mice models for anti-fungal treatment. In mice, posaconazole treatment at 20mg/kg body weight does not induce adrenal insufficiency and does not change the systemic level of steroids. Therefore, in human it might be possible to keep the dose at certain level that would only inhibit immune cell steroidogenesis (which is weaker than adrenal steroidogenesis). But all these need further research which are out of the scope of this manuscript.
4.19	Discussion - Line 377 – your statement relating to ref 54 is incorrect. This study is about antifungal prophylaxis not treatment, and while posaconazole reduced the rate of invasive fungal infections it did not improve survival in the overall population	Thank you. We have corrected this mistake in the revised manuscript.

4.20	- I feel that the information about steroid biosynthesis within immune cells is interesting and potentially targetable, but I don't think that this is adequately linked with the effects of posaconazole, nor do I think that these experiments justify a clinical trial of posaconazole in TNBC	A patent application is pending, and we are initiating a multicentre Phase II clinical trial on posaconazole's use in TNBC. The trial has already received approval from the ethics committee at the lead centre (Ethics Approval No.: SDTHEC2024002011), and we are currently in the process of registering the clinical trial. We cannot exclude the possibility of off-target effects, however, only the ongoing clinical trial and further research can resolve these questions. In this manuscript we want to restrict its successful use in tumour restriction data.
------	--	--

Response to the reviewers

We thank all the reviewers for their valid questions, constructive comments, and suggestions, which have helped us improve this manuscript. We are pleased that you found our work interesting and important. Many thanks for recommending our manuscript for acceptance.

Below, we provide a point-by-point responses to the referee's (Reviewer 5) comments.

Reviewer 5

Reviewer: "I do have additional questions and comments regarding the potential clinical application of drugs inhibiting steroidogenesis, including posaconazole. Does the planned phase II clinical trial involve administering posaconazole in a neoadjuvant setting? Most initial phase II trials are designed in the metastatic setting, where the patients enrolled will have received previous treatment, especially if it is for TNBC. Because the TNBC samples used in this study were from treatment-naive patients, the TME, including the immunosuppressive environment and steroidogenesis, may not reflect the heavily treated or actively progressive tumors in the metastatic setting. I suggest a careful statement in the summary paragraph regarding the "clinical trial phase" or "optimistic" precise and effective treatment, which cannot be concluded only from a planned phase II trial that has not been registered yet."

Authors: We appreciate reviewer's comment and thankful for their constructive suggestion. Most Phase II trials for drugs are initially designed for the advanced (Stage IV, metastatic) setting, where patients often haven't responded to standard treatments and have shorter progression-free survival (PFS), increasing the likelihood of positive results.

We have actually planned two Phase II clinical trials:

1. **Early Neoadjuvant Setting** (primary trial) – comparing chemotherapy + PD-1 inhibitor with or without posaconazole.
2. **Advanced (Stage IV, Metastatic) Setting** – comparing physician's choice of treatment (TPC) with or without posaconazole.

The reasoning behind this dual approach is:

- **Safety:** Posaconazole, a second-generation triazole antifungal drug, is already widely used in lymphoma and haematological oncology to prevent and treat fungal infections. Its safety profile has been confirmed in cancer patients, supporting its use in Phase II trials.

- **Efficacy:** The drug has shown efficacy in animal models, and its mechanism supports its use for both early and late-stage TNBC based on the database.
- **Why Focus on Early Neoadjuvant?** Posaconazole mainly boosts anti-tumour immune responses. Breast cancer is considered an “immunologically cold tumour,” and current guidelines recommend PD-1/PD-L1 inhibitors in both early neoadjuvant settings and advanced first-line treatments. Immunotherapy tends to be less effective in the late-line setting for two main reasons:
 - a. Late-stage patients often have compromised immune systems, making immunotherapy less effective.
 - b. Immunotherapy works best when the primary tumour or lymph nodes are still present, as T cell diversity and number decrease once the primary tumour is removed. Applying immunotherapy in the neoadjuvant stage, when the tumour hasn’t been excised, can induce more T cells that continue to target micro-metastatic cells, improving outcomes.”

We confirm, the clinical trial is already registered with NCI: [NCT06802757](https://clinicaltrials.gov/ct2/show/study/NCT06802757).

Reviewer: An important biomarker in TNBC that reflects subtypes within TNBC is the androgen receptor. Despite the study's main findings regarding corticosteroid hormones, the results on Androgens in Figure 1D about the integrated metabolomics data on steroid hormone concentrations with transcriptomic data on genes and receptors related to steroids are intriguing. It would be interesting to segregate the clinically AR-positive and AR-negative samples for analysis and observe the overall score to confirm that corticosteroids have higher scores.

Authors: Thank you for your insightful comment regarding the androgen receptor as an important biomarker in TNBC subtypes. We appreciate your suggestion to segregate clinically AR-positive and AR-negative samples for analysis and observe the overall score to confirm higher corticosteroid scores. We have now incorporated this as a promising future direction in our discussion, emphasizing the need for an in-depth investigation to validate these findings and further elucidate the relationship between AR status and corticosteroid levels in TNBC.

In the revised manuscript, the text will appear as “It would be valuable to segregate clinically AR-positive and AR-negative TNBC samples to analyse and compare their overall scores, potentially confirming higher corticosteroid levels and further elucidating the relationship between AR status and steroid hormone profiles in TNBC subtypes.”

Reviewer: I have some minor comments about some of the editing errors noted in the revised portions of the manuscript.

Authors: Thanks for your time with careful and detailed comments. We appreciate your suggestions.

Reviewer: 1. pg 4-5. lines 114-127. Glucocorticoids are used instead of GC, and I think you are referring to glucocorticoid receptor with GR? It has not been abbreviated before this paragraph. Also, there are numerous places throughout the manuscript where glucocorticoid is used instead of GC.

Authors: We agree and corrected as suggested.

2. pg. 7. lines 209-210. The sentence about Figure S2B is unnecessary, as is the repeated figure in Figures 2D and S2B.

Authors: Sorry for the confusion. Figure 2B is meaningful to show the steroid production by cancer cells and it was a reviewer's suggestion to test this and include.

3. pg. 10. lines 308-311. The added explanation about posaconazole alters the wording about the posaconazole Cyp11a1 activity. I suggest rewriting the sentence for clarity.

Authors: Thank you and sorry for the error. We have now corrected this as suggested.

4. Figure 5D needs labeling or explanation about the two columns.

Authors: Sorry for this mistake. Corrected as suggested. Thank you.

5. I consider posaconazole a generic name and thus would not use capital letters throughout the manuscript. It is currently used in both uppercase and lowercase letters.

Authors: Thank you for notifying this. Corrected as suggested.